# Hybrid Nanoparticles Based on Cobalt Ferrite and Gold: Preparation and Characterization

**Svetlana Saikova [1,2,\*], Alexander Pavlikov [1], Tatyana Trofimova [1], Yuri Mikhlin [2], Denis Karpov [1], Anastasiya Asanova [1,3], Yuri Grigoriev [1], Mikhail Volochaev [4], Alexander Samoilo [1], Sergey Zharkov [1,4] and Dmitry Velikanov [4]**

[1] School of Non-Ferrous Metals and Material Science, Siberian Federal University, 660041 Krasnoyarsk, Russia; hahanka@yandex.ru (A.P.); ttv91@mail.ru (T.T.); denikarp@mail.ru (D.K.); nastasia.asanova@gmail.com (A.A.); gr2897@gmail.com (Y.G.); x-lab@rambler.ru (A.S.); zharkov@iph.krasn.ru (S.Z.)

[2] Institute of Chemistry and Chemical Technology, Federal Research Center "Krasnoyarsk Science Center of the Siberian Branch of the Russian Academy of Sciences", 660036 Krasnoyarsk, Russia; yumikh@icct.ru

[3] Institute of Agroecological Technologies, Krasnoyarsk State Agrarian University, 660049 Krasnoyarsk, Russia

[4] Kirensky Institute of Physics, Federal Research Center "Krasnoyarsk Science Center of the Siberian Branch of the Russian Academy of Sciences", Akademgorodok, 660036 Krasnoyarsk, Russia; volochaev91@mail.ru (M.V.); dpona1@gmail.com (D.V.)

\* Correspondence: ssai@mail.ru; Tel.: +7-(902)-991-20-31

**Abstract:** During the past few decades, hybrid nanoparticles (HNPs) based on a magnetic material and gold have attracted interest for applications in catalysis, diagnostics and nanomedicine. In this paper, magnetic $CoFe_2O_4$/Au HNPs with an average particle size of 20 nm, decorated with 2 nm gold clusters, were prepared using methionine as a reducer and an anchor between $CoFe_2O_4$ and gold. The methionine was used to grow the Au clusters to a solid gold shell (up to 10 gold deposition cycles). The obtained nanoparticles (NPs) were studied by X-Ray diffraction (XRD), transmission electron microscopy (TEM), Fourier-transform infrared (FT-IR) spectroscopy, X-Ray photoelectron spectroscopy (XPS) and UV-vis spectroscopy techniques. The TEM images of the obtained HNPs showed that the surface of cobalt ferrite was covered with gold nanoclusters, the size of which slightly increased with an increase in the number of gold deposition cycles (from 2.12 ± 0.15 nm after 1 cycle to 2.46 ± 0.13 nm after 10 cycles). The density of the Au clusters on the cobalt ferrite surface insignificantly decreased during repeated stages of gold deposition: 21.4 ± 2.7 Au NPs/$CoFe_2O_4$ NP after 1 cycle, 19.0 ± 1.2 after 6 cycles and 18.0 ± 1.4 after 10 cycles. The magnetic measurements showed that the obtained HNPs possessed typical ferrimagnetic behavior, which corresponds to that of $CoFe_2O_4$ nanoparticles. The toxicity evaluation of the synthesized HNPs on *Chlorella vulgaris* indicated that they can be applied to biomedical applications such as magnetic hyperthermia, photothermal therapy, drug delivery, bioimaging and biosensing.

**Keywords:** $CoFe_2O_4$/Au nanoparticles; toxicity; X-ray photoelectron spectroscopy; anion-exchange resin precipitation; hybrid nanoparticles; synthesis; magnetic hysteresis loops

## 1. Introduction

The design, synthesis and applications of hybrid nanoparticles (HNPs) are hot research topics in the field of nanotechnology. HNPs are important due to their multi-functionalities, which are difficult to achieve by a simple mixture of individual components. Due to the combination of physical and chemical (magnetic and optical) properties of compound substances, they can be used in various fields, such as chemistry, physics, microelectronics, biotechnology, catalysis and medicine [1–5].

Recently, the use of magnetic HNPs in theranostics (combining therapeutic and diagnostic functions in one nano-object) has remarkably increased [6–9]. When an external magnetic field is applied to magnetic nanoparticles, they can be directed to a specific area (organ or tissue) along with drug molecules attached to their surfaces [10–13].

Among various iron oxide-based nanomaterials, magnetite has been the most popular for biomedical applications. Additionally, it has the strongest magnetism among iron oxides. However, magnetite nanoparticles exhibited toxicity in cell cultures and animal studies [14,15]. Additionally, magnetite is extremely sensitive to oxidation, decreasing its magnetic properties; thus, it requires a protective capping layer [16,17]. Cobalt and nickel ferrites are often considered as alternatives to magnetite [18,19]. These materials are attractive due to their high chemical stability and significant magnetism.

Due to recent advances in surface chemistry, nanoparticle functionalization has been tremendously developed. Gold is one of the most effective coating materials to protect magnetic nanoparticles and increase their biocompatibility. Additionally, $CoFe_2O_4$/Au systems are of interest for catalytic applications, combining recovery and reuse, as well as increased activity under illumination. Such a hybrid catalyst material has higher efficiency and lower cost due to the larger Au surface area, compared with gold NPs. It is also worth noting that it is possible to separate the catalyst using an external magnetic field [4,5,20–26]. The catalytic properties of gold-coated magnetic nanoparticles can be varied by the type of coating (thick shell or decoration with nanoclusters) and the morphology and size of gold nanoparticles on the surface.

While numerous studies have demonstrated the production of hybrid gold-based nanoparticles, a reliable and predictable synthetic strategy that provides reproducibility has not been realized. There are two main approaches in the literature to synthesize these materials: In the first case, gold seeds with a diameter of 1–3 nm are prepared by the reduction with borohydride [27] or sodium citrate [28]. Then, the obtained seeds are added to magnetic particles. The surface of the particles is pre-modified with biomolecules or polymers to attach gold nanoclusters. Polyethyleneimine (PEI) is a well-established agent for the surface treatment of magnetic nanoparticles [29,30]. Another technique involves creating an intermediate layer of silica dioxide with grafted amino groups of 3-aminopropyltrimethoxysilane (APTMS) or mercapto groups on the surface of magnetic nanoparticles [31,32]. However, it is often challenging to attach gold seeds onto the nanoparticle surface due to the large lattice mismatch between the magnetic material and gold.

Another approach is to form a surface complex with gold ions. The surface of the magnetic material is modified with bifunctional ligands, such as citrate [33,34], dithiosuccinic acid [5] and methionine [35]. The bifunctional ligand must have a high affinity for both the magnetic nanoparticle and gold ions. Additionally, an excess ligand can act as a reducing agent. Additional reducing agents (ascorbic acid [36,37] and hydroxylamine [38]) may be used in this approach. However, gold is often not deposited on the NP surface, but forms large particles (50–80 nm) in the solution. To control gold deposition, a reducing agent, modifier and stabilizer must be selected carefully.

We provide a simple, fast and easily reproducible method to obtain cobalt ferrite HNPs, where a strong-base anion-exchange resin in OH form is used as a precipitant of cobalt and ferric ions. During the process, anions in the solution ($Cl^-$) exchange with the $OH^-$ ions of the resin (the $Co^{2+}$, $Fe^{3+}$ and $OH^-$ given off by the resin can combine to form hydroxides). This process is conducted under easily controllable stationary conditions and allows obtaining homogeneous nanoparticles with uniform size and morphology without impurity ions. This method eliminates the use of expensive equipment and multiple washing steps [39,40]. The anion-exchange resin precipitation is conducted in the solution containing dextran-40 to prevent the aggregation of the as-obtained magnetic nanoparticles and control the particle size. As we have previously discussed [41], the formation of new nuclei dominates crystal growth due to steric stabilization as a consequence of polymer adsorption on the surface of magnetic nanoparticles. In this

study, we used L-methionine as a reducing agent, an anchor between $CoFe_2O_4$ and gold and a stabilizing agent for nanoparticles. One of the aims of the present research was to test the hypothesis about the possibility that the further growth of the Au clusters to a solid shell is caused by methionine multi-stage gold reduction [35]. Additionally, we aimed to examine what changes in the state of the system are produced with an increase in the gold deposition stages. To the best of our knowledge, ours is the first article examining the effect of multistage gold depositions on the $CoFe_2O_4$ nanoparticles surface coverage.

## 2. Materials and Methods

### 2.1. Chemicals

Cobalt chloride ($CoCl_2 \cdot 6H_2O$), iron chloride ($FeCl_3 \cdot 6H_2O$), tetrachloroauric(III) acid ($HAuCl_4$), L-methionine ($CH_3SCH_2CH_2CH(NH_2)CO_2H$), dextran (($C_6H_{10}O_5$)n Mr ~40,000 g/mol) and other chemicals were of analytical grade, were purchased from Sigma-Aldrich and were used as received. The strong-base anion-exchange resin AV-17-8 with a polystyrene gel matrix was produced by "Azot" Corporation (Cherkassy, Ukraine) in the chloride form with a bead size of 0.4–0.6 mm (Russian GOST 20301-74). This resin has a gel matrix, based on polystyrene cross-linked with divinylbenzene and the functional group quaternary ammonium (type I). This resin is an analogue of Purolite A400/A300, Lewatit M-500, Amberlite IRA 402/420, Dowex SBR-P/ Maraton A; it has a significantly lower cost and is widely used in separation, purification and decontamination processes in Russia.

### 2.2. Synthesis of Cobalt Ferrite Nanoparticles

The proposed approach differs from the reported one [42] in nature of magnetic NPs, in the technique of their production, in the concentration of the aqueous solution applied and in their synthesis modes. In this study, cobalt ferrite nanoparticles were synthesized by anion-exchange resin precipitation. The synthesis procedure was described in detail in [41]. Briefly, $CoCl_2 \cdot 6H_2O$ (1.2 g) and $FeCl_3 \cdot 6H_2O$ (3.4 g) were dissolved in dextran-40 (50 mL, 10%) to stabilize the colloidal system and reduce the particle size. Excess AV-17-8(OH) (150%) was added to a solution containing cobalt and ferric salts. The mixture was stirred (180 rpm) at a temperature of 60 °C, using a magnetic stirrer, for 1 h. To remove the anion-exchange resin beads from the reaction products, a sieve with round holes (0.16 mm in diameter) was used; the precipitate was centrifuged, washed with distilled water, dried in air at 80 °C and then annealed in a muffle furnace at 600 °C, for 1 h.

### 2.3. Synthesis of Hybrid $CoFe_2O_4$/Au Nanoparticles

Hybrid $CoFe_2O_4$/Au nanoparticles were fabricated by a methionine-directed gold deposition from $HAuCl_4$ solution, according to a modified technique described in [35]. Cobalt ferrite nanoparticles (25 mg) were dispersed in L-methionine (20 mL, 0.03 mol/L) by ultrasonic agitation (ultrasonic bath "sapphire"), for 30 min. Chlorauric acid (20 mL, 0.6 mmol/L) was added to the mixture; pH was adjusted to ten by the addition of NaOH (1 mol/L). The deposition process was conducted at 37 °C for 4 h under stirring (800 rpm). The resulting nanoparticles were separated by magnetic separation and thoroughly washed with distilled water and ethanol. The procedure of gold deposition was repeated up to ten times under the same conditions. The nanoparticles after one, six and ten deposition cycles were collected for characterization.

### 2.4. Nanoparticle Characterization

The phase composition of samples was determined on a Shimadzu XRD-6000 (Shimadzu Corporation, Kyoto, Japan) diffractometer employing monochromatic CuK$\alpha$ radiation; phase identification was performed according to the JCPDS powder diffraction

file №22-1086. TEM, EDX analysis and selected area electron diffraction characterization were carried out using a Hitachi 7700M (Hitachi Corporation, Hitachi, Japan, the accelerating voltage: 110 kV) and a JEM 2100 (JEOL, Tokyo, Japan) transmission electron microscope operated at an accelerating voltage of 200 kV. A cobalt ferrite particle size distribution histogram was obtained from more than 300 particles. UV-vis absorption spectra were collected in a glass cell with an optical path of 1 cm, employing a GENESYS 10S UV-Vis spectrophotometer (Thermo Scientific, Bedford MA, USA). The FTIR spectra of samples were recorded on a Tensor 27 (Bruker, Germany) FTIR spectrometer in the range of 4000–400 $cm^{-1}$. XPS studies were performed using a hydrosol, dried with highly oriented pyrolytic graphite (HOPG) and gently rinsed with water. The spectra were acquired using a SPECS spectrometer (SPECS Gmbh, Berlin, Germany) equipped with a PHOIBOS 150-MCD-9 hemispherical electron analyzer. Spectra were recorded upon excitation with a monochromatic radiation of AlK$\alpha$ (E = 1486.6 eV). The analyzer pass energy was 10 eV for high-resolution scans and 20 eV for survey spectra. An electron flood gun was applied to eliminate inhomogeneous electrostatic charging of the samples; the C 1s peak at 284.45 eV from HOPG was used as a reference. The high-resolution spectra were fitted after the subtraction of Shirley-type background with Gaussian–Lorentzian peak profiles using CasaXPS software (version 2.3.16, Casa Software, Teignmouth, UK). The magnetic properties of the material obtained were investigated using a vibrating sample magnetometer, in a magnetic field up to ±10 kOe at 298 K [43]. An electromagnet with high magnetic field uniformity was used as a source. The magnetic measurements were performed using a direct method of measuring the inductive electromotive force. The mechanical vibrations of the sample were provided by a vibrator of the original design [44]. The relative instability of the oscillation amplitude was 0.01%, with a frequency of 0.001%. The registration of the signal was conducted using the system of four pickup coils. The dynamic range of the device was $5 \cdot 10^{-6}$–$10^2$ emu. The toxicity of the nanoparticles was studied on an algologically clean test culture of *Chlorella vulgaris* Beijer in the exponential stage of growth. The experiment was carried out in a multi-cell cultivator, KVM-05 (Europolytest, Moscow, Russia), in four parallels. The microalgae species were cultivated at a medium temperature of 36 °C, light irradiation with an average intensity of 60 $W/m^2$ and $CO_2$ provision from the air (0.03%). 24 samples of microalgae species in 2% nutrient medium of Tamiya (6 $cm^3$) were placed into cuvettes (10 $cm^3$). Nanoparticles were introduced into the cultivation water, in concentrations of 0.08, 0.46, 2.78, 3.13, 6.25, 12.50, 16.67 and 25 mg/L. The viability of the test culture (Equation (1)) in the sample medium was determined by the change in optical density relative to the reference using the spectrometer IPS-03 (Russia, Europolytest) Equation (1):

$$\text{Viability} = (1 - (D_R - D_{EXP})/D_R) \cdot 100\%, \qquad (1)$$

where $D_R$ and $D_{EXP}$ are the average values of optical density in the reference and the experiment, respectively.

### 3. Results

Cobalt ferrite nanoparticles were synthesized using anion-exchange resin precipitation based on the ion exchange between the anions of the aqueous solution and a solid substance (an anion-exchange resin, containing the OH groups). The co-precipitation of $Co(OH)_2$ and $Fe(OH)_3$ can be described by Equation (2):

$$5R\text{-}OH + CoCl_2 + FeCl_3 \rightarrow 5R\text{-}Cl + Co(OH)_2\downarrow + Fe(OH)_3\downarrow, \qquad (2)$$

where R-OH, R-Cl—anion-exchange resin AV-17-8 with OH- and Cl- groups on the resin substrate, respectively.

The precipitation of metal hydroxides occurs at the resin–solution interface, namely on the resin beads. When the thickness of the surface deposit reaches 1–1.5 μm, it is exfoliated, and an individual product phase forms [39,40]. We used a polysaccharide, dextran-40, to obtain homogeneous nanoparticles with similar sizes and morphologies. Ac-

cording to [41], dextran-40 adsorbs on the primary hydroxide particles and protects against coalescence and aggregation.

Thus, the precipitate obtained is free from impurity ions because no additional reagents besides the anion-exchange resin are used, and the anions of the initial salts are trapped by polymer beads. Accordingly, the product does not require multiple washing steps. The product annealed for 1 h at 600 °C was investigated by X-Ray diffraction (XRD) (Figure 1a). The position and relative intensity of all diffraction peaks match well with the standard $CoFe_2O_4$ diffraction data JCPDS №22-1086 (element concentrations, wt.%: Co-25.1, Fe-47.6, O-27.2). The average crystallite size determined from the peak broadening for the four most intense XRD peaks using the Scherrer equation (Equation (3)) was 12.4 ± 1.0 nm. This result is in a good agreement with transmission electron microscopy (TEM) characterization, which shows that the obtained nanoparticles have a mean size 11.8 ± 2.3 nm.

$$D = (b·\lambda)/(FWHM·cos\Theta) \tag{3}$$

where FWHM is the full width at the half-maximum intensity of the peak, $2\theta$ is the scattering angle in radians, $\lambda$ is the wavelength (1.5406 Å), b is a constant, which is a function of the crystallite geometry, and is taken between 0.89 and 0.94, and D is the average size of the crystallites.

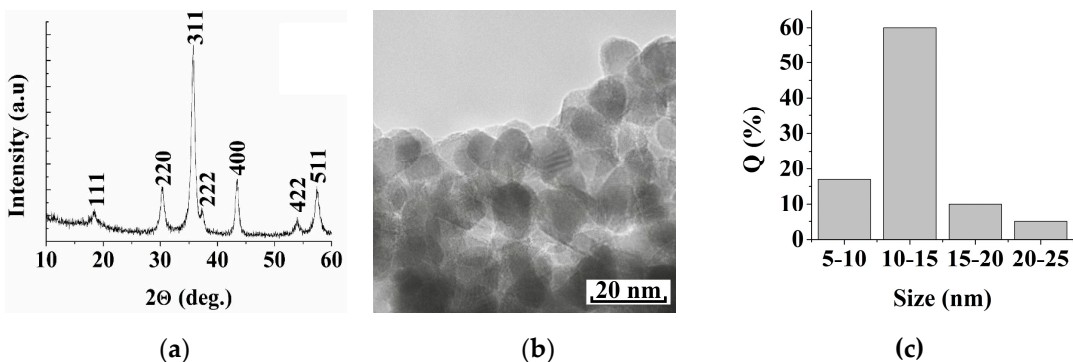

**Figure 1.** (**a**) XRD–pattern, (**b**) TEM image and (**c**) size distribution histogram of cobalt ferrite particles synthesized by anion-exchange resin co-precipitation in a solution of $FeCl_3$, $CoCl_2$ and 10% dextran–40 at 60 °C and annealed at 600 °C for 1 h.

To obtain hybrid nanoparticles based on cobalt ferrite and gold, we modified the technique reported in [35]. This technique includes several stages. The first step was to treat cobalt ferrite nanoparticles by L-methionine. According the literature [35], the amino acid particle surface modification facilitates the attachment of gold nanoclusters. Then, the hydrogen tetrachloroaurate (III) and NaOH solutions were added. As a result, the seeds of $Au^0$ were deposited on the ferrite nanoparticle surface during reduction by L-methionine, without using other reducing agents. The L-methionine and $H[AuCl_4]$ treatments of the nanoparticles were carried out in 1, 6 and 10 deposition cycles.

We used FTIR spectroscopy to determine the presence of methionine on the surface of magnetic nuclei after L-methionine pre-treatment and the nature of the bonds between the amino acid and $CoFe_2O_4$ nanoparticles surface. Figure 2 shows the FTIR absorption spectra of the cobalt ferrite nanoparticles before (curve 1) and after (curve 2) their treatment by L-methionine (Figure 2b shows the enlarged spectra of $CoFe_2O_4$ and $CoFe_2O_4$/Au NPs in the region 1800–1200$^{-1}$ cm$^{-1}$). The spectra of both samples were similar and contained the absorption bands at 590 and 450 cm$^{-1}$, which are typical for cobalt ferrite and correspond to the stretching vibrations of metal-oxygen bonds. The weak absorption bands at 3400 cm$^{-1}$ was due to the stretching vibrations of OH groups and the deformation vibrations of water molecules adsorbed on the surface of $CoFe_2O_4$ nanoparticles. In the range of 1800–1200 cm$^{-1}$ of the spectrum of sample 2, a few weak lines corre-

sponding to the deformation vibrations of $NH_3^+$ (1508 cm$^{-1}$), asymmetric and symmetric vibrations of COO$^-$ groups (1581 and 1402 cm$^{-1}$), scissoring vibration $H_2O$ (1622 cm$^{-1}$) and deformation vibration of $CH_3$ groups (1346 cm$^{-1}$) of methionine were observed [45]. The low intensities of these peaks indicate the weak adsorption of methionine on the surface of cobalt ferrite nanoparticles after one cycle of treatment by L-methionine. The frequencies of $\nu s$(COO$^-$) and $\delta$($NH_3^+$) downshifted slightly (from 1414 to 1402 cm$^{-1}$ and from 1515 to 1508 cm$^{-1}$, respectively) upon the treatment of ferrite NPs with methionine molecules. Such frequency downshifts may be due to the interaction of COO$^-$ and $NH_3^+$ groups of the amino acid with the surface of cobalt ferrite nanoparticles [35]. Our findings confirmed the presence of methionine on the surface of $CoFe_2O_4$. However, due to the low intensity of the spectrum in this area, it was difficult to draw unambiguous conclusions about the nature of the bonds between the amino acid and $CoFe_2O_4$ nanoparticles surface.

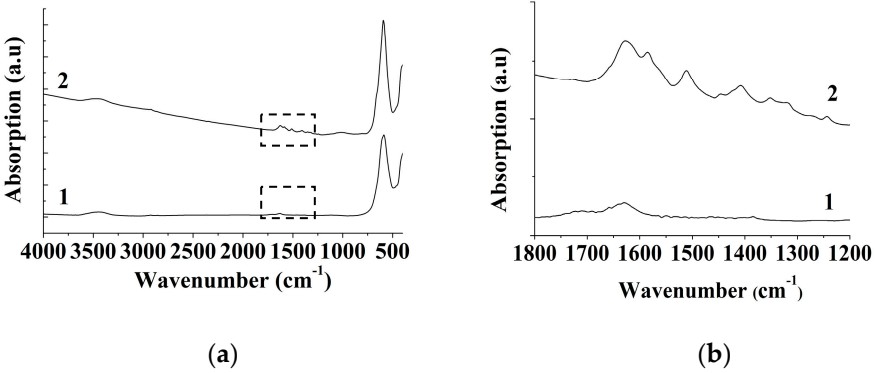

(**a**) (**b**)

**Figure 2.** FT-IR-spectra of cobalt ferrite nanoparticles in the spectral region of 4000−500 cm$^{-1}$ (**a**) and enlarged spectra in the region of 1800−1200 cm$^{-1}$ (highlighted in the figure a with a rectangle) (**b**) synthesized by anion-exchange resin co-precipitation in a solution of $FeCl_3$, $CoCl_2$ and 10% dextran-40 at 60 °C and annealed at 600 °C for 1 h before (1) and after (2) treatment by methionine (20 mL, 0.03 mol/L).

Figure 3 shows the TEM images of hybrid $CoFe_2O_4$/Au nanoparticles after 1, 6 and 10 cycles of gold deposition. We can see small gold clusters on the cobalt ferrite nanoparticles, as confirmed by the electron microdiffraction analysis (Figure 3d–f), the energy-dispersive X-ray spectroscopy (Figure 4) and the X-Ray diffraction analysis (Figure 5). It should be noted that gold deposition does not affect the structure and stoichiometry of cobalt ferrite (element concentrations, wt.%: Co-24.3, Fe-46.0, O-26.3, Au-3.3). The size of the gold nanoclusters slightly increased with an increase in the number of gold deposition cycles: 2.12 ± 0.15 nm for 1 cycle, 2.15 ± 0.15 nm for 6 cycles and 2.46 ± 0.13 nm for 10 cycles. However, we noticed that the density of the gold nanoclusters on the cobalt ferrite surface does slightly decrease with the increase in the number of gold deposition cycles: 21.4 ± 2.7 Au NPs/$CoFe_2O_4$ NP after 1 cycle, 19.0 ± 1.2 after 6 cycles and 18.0 ± 1.4 after 10 cycles.

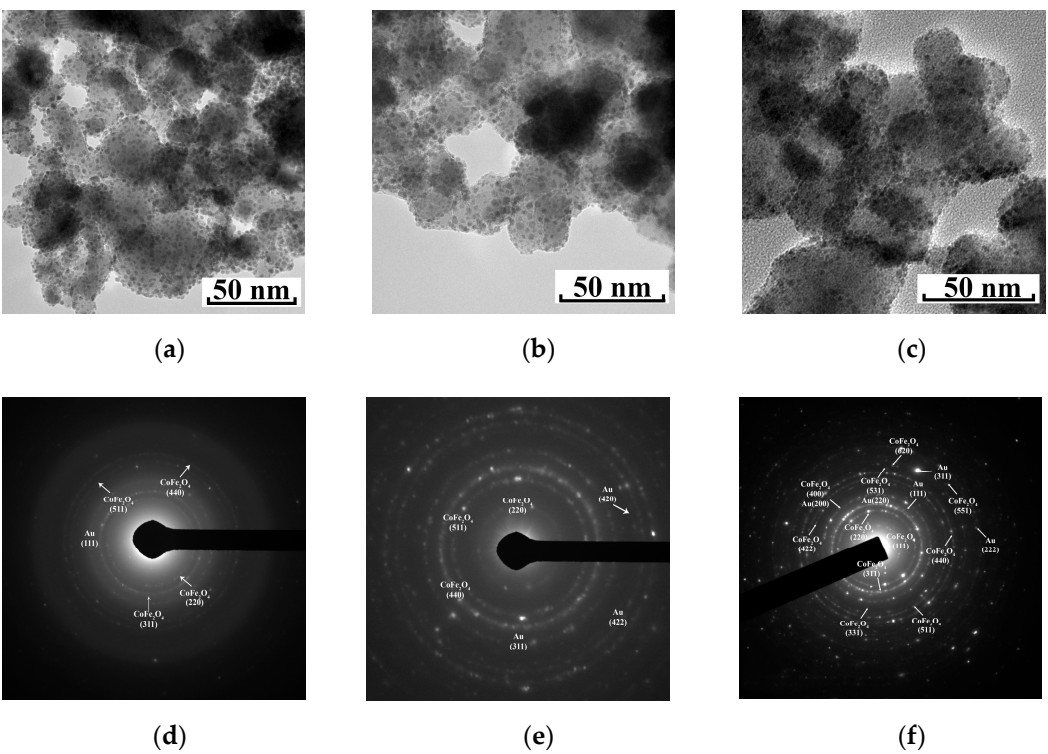

**Figure 3.** TEM images (**a**–**c**) and electron diffraction patterns (**d**–**f**) of CoFe₂O₄/Au nanoparticles after gold deposition from the alkaline (pH = 12) solution containing 0.03 mol/L HAuCl₄ and 0.03 mol/L methionine at 37 °C for 4 h: (**a**,**d**) 1 gold deposition cycle, (**b**,**e**) 6 gold deposition cycles and (**c**,**f**) 10 gold deposition cycles.

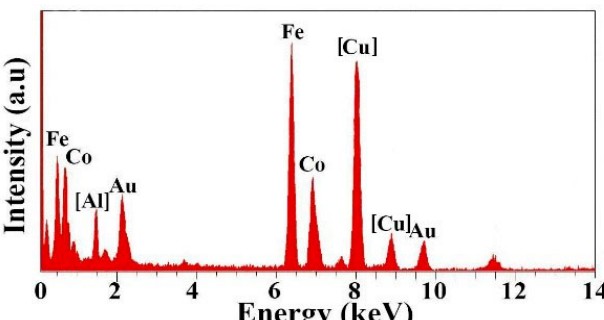

**Figure 4.** Energy-dispersive X-ray spectrum of CoFe₂O₄/Au nanoparticles after 6 gold deposition cycles. Al and Cu signals are put it in brackets since their source is the sample holder for EDX analysis.

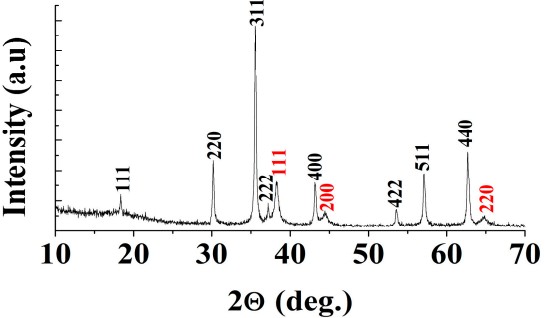

**Figure 5.** XRD pattern of CoFe₂O₄/Au nanoparticles after 10 gold deposition cycles from the alkaline (pH = 12) solution containing 0.03 mol/L HAuCl₄ and 0.03 mol/L methionine at 37 °C for 4 h:

the reflexes corresponding to $CoFe_2O_4$ are marked in black, and the reflexes corresponding to Au are marked in red.

The obtained particles after 1 gold deposition cycle redispersed in ethanol and the solution remaining after the separation of the magnetic particles were studied by a spectrophotometer in the wavelength range of 400–1000 nm. The dispersed $CoFe_2O_4$/Au NPs show an absorption peak at 580 nm (Figure 6). The shift of the characteristic peak for Au NPs (520 nm) is a consequence of their adsorption on the surface of cobalt ferrite nanoparticles [46]. The optical density of the solution after the magnetic separation of nanoparticles is much lower than in the initial hydrosol and may be due to the residual amounts of nanoparticle. This indicates that the reduction of gold occurs on the surface of the cobalt ferrite and not in the volume of the solution.

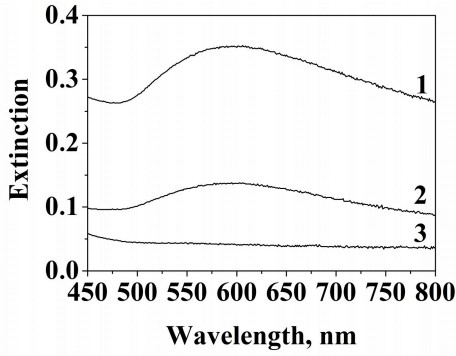

**Figure 6.** Optical absorption spectra of $CoFe_2O_4$ nanoparticles before (3) and after (1) 1 gold deposition cycle from the alkaline (pH = 12) solution containing 0.6 mmol/L $HAuCl_4$ and 0.03 mol/L methionine at 37 °C for 4 h, and spectra of solution (2) remaining after $CoFe_2O_4$/Au nanoparticles (NPs) magnetic separation.

X-ray photoelectron spectroscopy (XPS) was applied to study the surface composition and chemical state of elements in the hybrid nanoparticles $CoFe_2O_4$/Au obtained after 1, 6, and 10 gold deposition cycles. The survey spectra (Figure 7) contain the peaks Au 4f, S 2p, O 1s, N 1s, C 1s, Fe 2p and Co 2p. The surface chemical composition of the $CoFe_2O_4$/Au NPs is presented in Table 1. (Concentrations of carbon originating from adsorbed ligands, HOPG support and adventitious carbon-bearing contaminations are omitted for clarity).

**Table 1.** Surface composition of $CoFe_2O_4$/Au NPs derived from XPS.

| Element | Concentration, at. % | | |
| | Number of Gold Deposition Cycles | | |
| | 1 | 6 | 10 |
|---|---|---|---|
| Au | 13.1 | 9.6 | 8.7 |
| N | 9.8 | 6.8 | 6.6 |
| O | 61.8 | 61.6 | 69.0 |
| Fe | 2.0 | 6.4 | 3.4 |
| Co | 3.3 | 3.8 | 3.2 |
| S | 10.0 | 9.8 | 9.1 |

According to the XPS data, the concentration of gold on the NPs surface decreases with an increase in the number of deposition cycles. After the first gold deposition cycle, metallic Au fixed on the surface of the $CoFe_2O_4$ particles (Table 2). Moreover, there were adsorbed $Au^+$ and $Au^{3+}$ ions in $CoFe_2O_4$/Au NPs, which desorbed during the further cycles of gold deposition (after treatment with methionine and other reagents, washing with water, a magnetic separation and ultrasonic treatment). However, $Au^0$ nanoparticles

were fixed on the NPs and only slightly detached from the magnetic core, under the influence of a magnetic field, the repetitive deposition cycles and during post-synthetic processing.

**Table 2.** Relative concentrations of Au species (%) and binding energies of the Au $4f_{7/2}$ peaks derived from XPS analysis.

| The Number of Gold Deposition Cycles | Au(0) | | Au(I) | | Au(III) | |
|---|---|---|---|---|---|---|
| | % | BE (eV) | % | BE (eV) | % | BE (eV) |
| 1 | 14.6 | 84.0 | 78.2 | $85.0 \pm 0.1$ | 7.2 | 85.8 |
| 6 | 76.8 | 83.9 | 23.2 | $85.0 \pm 0.1$ | - | - |
| 10 | 76.6 | 84.0 | 23.4 | $85.0 \pm 0.1$ | - | - |

The N/S atomic ratios obtained from XPS spectra of all the samples ranged from 0.8 to 0.9. This indicates the presence on the surface of the obtained nanoparticles adsorbed methionine as well as the product of its oxidation by $Au^{3+}$, which is believed to be methionine sulfoxide $CH_3$-S(O)$CH_2CH_2$C($NH_2$)COOH [47].

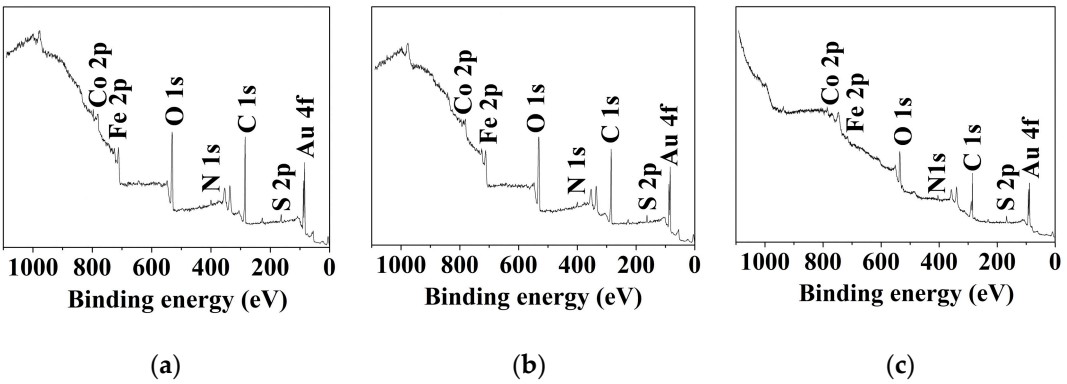

**Figure 7.** Wide XPS spectra of CoFe$_2$O$_4$/Au synthesized under the conditions in Figure 3: (**a**) 1 gold deposition cycle, (**b**) 6 gold deposition cycles and (**c**) 10 gold deposition cycles.

To gain a better insight into the chemical state of sulfur at the surface of CoFe$_2$O$_4$/Au, the narrow XPS spectra for S 2p were recorded (Figure 8). The spectrum S 2p (Figure 8a) of the hybrid nanoparticles was obtained after one cycle of gold deposition showed a doublet with a binding energy S $2p_{3/2}$ 162.6 eV. The energies of the peaks in the spectrum are similar to those of methionine [42] and can be assigned to the thioether group C-S-C [48,49].

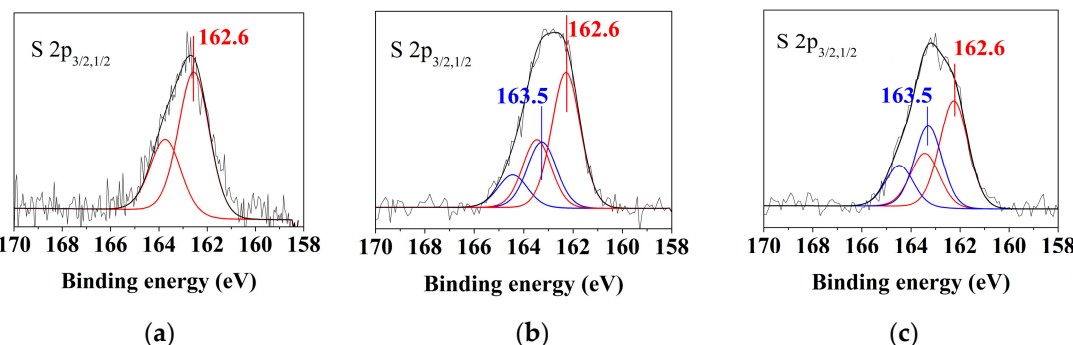

**Figure 8.** Photoelectron S 2p spectra of CoFe$_2$O$_4$/Au nanoparticles synthesized under the conditions in Figure 3: (**a**) 1 gold deposition cycle, (**b**) 6 gold deposition cycles and (**c**) 10 gold deposition cycles.

As the number of the gold deposition cycles increase (Figure 8b,c) the major doublet S 2p does not change their position; however, a second minor doublet with the binding energy S $2p_{3/2}$ of 163.5 appears in the spectra. This doublet can be attributed to the product of methionine oxidation. The lines with $S2p_{3/2}$ binding energies of 163.5, which are typically assigned to polysulfide species, were observed in the XPS spectra of core-shell NiFe$_2$O$_4$@Au nanoparticles obtained by the reduction of gold with methionine [42]. These results indicate the dimerization of methionine during its oxidation with the formation of two-center three-electron (2c–3e) S–S bonds [50].

The spectra of Au $4f_{7/2,5/2}$ from CoFe$_2$O$_4$/Au nanoparticles, obtained after one cycle of gold deposition (Figure 9a), can be fitted to three components. The major doublet of the Au $4f_{7/2}$ peak, with a binding energy of 84.9 eV, corresponds to Au$^+$ species; the component at 84.0 eV is due to metallic Au, and the third doublet of Au $4f_{7/2}$ with a binding energy of 85.6 eV corresponds to Au(III). The presence of Au (III) in the spectra can be explained by unreacted residual reagent or by the fact that [AuCl$_2$]$^-$ can be disproportionate in aqueous solutions, forming [AuCl$_4$]$^-$ and metallic gold (Equation (4)).

$$3[AuCl_2]^- = 2Au^0 + [AuCl_4]^- + 2Cl^- \tag{4}$$

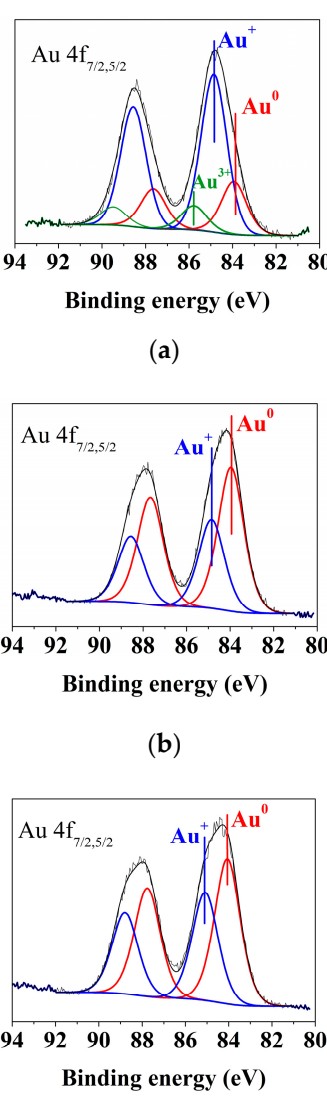

(**a**)

(**b**)

(**c**)

**Figure 9.** Au 4f spectra of CoFe$_2$O$_4$/Au nanoparticles synthesized under the conditions in Figure 3: (**a**) 1 gold deposition cycle, (**b**) 6 gold deposition cycles and (**c**) 10 gold deposition cycles.

The spectra of Au 4f of $CoFe_2O_4$/Au nanoparticles obtained after 6 and 10 cycles of gold deposition (Figure 9b,c) can be decomposed into two components: $Au^0$ and $Au^+$ with binding energies, given in Table 2. The increase in the number of deposition cycles from one to six leads to almost a twofold increase in $Au^0$ concentration and a twofold decrease in $Au^+$ concentration. If the number of cycles rises further, the concentrations of the Au species on the surface of the hybrid nanoparticles stay the same. The presence of Au (I) spectral lines can be explained by the adsorption of $[AuCl_2]^-$ or gold (I) sulphide on the surface of nanoparticles and should be investigated further.

Figure 10 shows the dependance of the magnetization of the nanoparticles before and after one cycle of gold deposition (the sample with the highest gold NPs concentration) on the applied magnetic field measured at 298 K of the sample with the highest gold nanoparticles concentration.

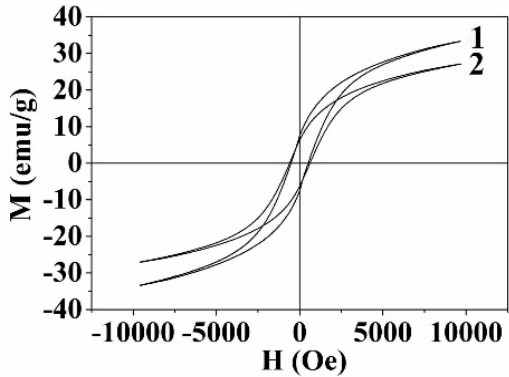

**Figure 10.** Magnetization curves of $CoFe_2O_4$ (1) before and (2) after one gold deposition cycle, measured in magnetic field H = ± 10 kOe at 298 K.

The values of the saturation magnetization (Ms) were estimated in magnetic field H = 10 kOe. The specific residual magnetizations (Mr) were determined from the point of intersection of the drop-down portion of the magnetization curve with the Y-axis. The values of coercivity (Hc) of the nanoparticles were determined by the measurement of the hysteresis loop width. The basic parameters of the hysteresis loops are collected in Table 3.

**Table 3.** Magnetic parameters obtained from the results of magnetic hysteresis loops in Figure 9.

| Sample | Ms, emu/g | Mr, emu/g | Hc, Oe |
|---|---|---|---|
| $CoFe_2O_4$ | 33.21 | 7.55 | 510 |
| $CoFe_2O_4$/Au | 27.04 | 6.31 | 600 |

The results show the ferrimagnetic behavior of both samples. The magnetic moments of various sublattices of ferrimagnetics, including cobalt ferrite, are oriented anti-paral -lel, but the magnitudes of their magnetic moments are not equal. As a result, the re- sulting magnetic moment is not equal to zero. The magnetic moments of octahedrally coordinated $Co^{2+}$ ions are uncompensated, whereas the magnetic moments of $Fe^{3+}$ ions are fully compensated due to the ionic spin of the octahedrally coordinated $Fe^{3+}$ ions being oriented opposite to the spins of the tetrahedrally oriented ions. In ferrimagnets, magnetic ordering is retained in the absence of an external field (the magnetizing effect).

The observed saturation magnetization values (33.21 emu/g $CoFe_2O_4$ and 27.04 emu/g for $CoFe_2O_4$/Au) were noticeably lower than the bulk values (80.8 emu/g) [51]. It is known that decreasing the particle size decreases the saturation magnetization of the material. This is generally attributed to the increase in surface spin canting caused by the structural disorder of surface atoms [52]. Decreasing the particle size leads to an increase in the relative number of atoms in the surface layer.

A decrease in the magnetic saturation and residual magnetization values per mass of the sample was observed for $CoFe_2O_4$/Au, mostly due to the presence of gold. According to thermogravimetric analysis (TGA) results (Supplementary Information, Figures S1 and S2), the amount of organic substances (amino acid methionine and its oxidation products) on the $CoFe_2O_4$/Au nanoparticles surface is negligible (2.5 wt.%). Thus, the organic material makes a small contribution to reducing the magnetic parameters. However, the coercive force of $CoFe_2O_4$/Au NPs increases slightly. We argue that this is primarily due to the loss of small gold-ferrite particles during additional washing and magnetic separation. The magnetic properties strongly depend on the fabrication procedure of the sample (annealing, preparation method, size and morphology of crystallites) [53]. The values of the magnetic parameters of $CoFe_2O_4$ nanoparticles reported in the literature vary from Ms~15–20 [54,55] to 60–70 emu/g [52,56].

The cytotoxicity of the obtained HNPs remains an important issue to realize its potential applications in biology and medicine. The unicellular green alga, *Chlorella vulgaris*, was used as a test organism in our experiments. This alga is a widely used test object; it can be grown in a synthetic medium and multiplies rapidly. The survival of the test organism depends on the concentration of nanoparticles for both $CoFe_2O_4$ and $CoFe_2O_4$/Au nanoparticles, obtained after one cycle of gold deposition. We found that gold-containing nanoparticles with a non-continuous gold shell showed lower toxicity compared with bare $CoFe_2O_4$ nanoparticles. At a nanoparticle concentration of 25 mg / L, the 85% viability of the test culture in the presence of $CoFe_2O_4$/Au was higher, compared with that (40%) in the case of bare $CoFe_2O_4$ nanoparticles (Figure 11). We speculate that the decrease in the toxicity of $CoFe_2O_4$/Au NPs also could be associated with the adsorption of a significant amount of amino acid L-methionine on the surface of the nanoparticles.

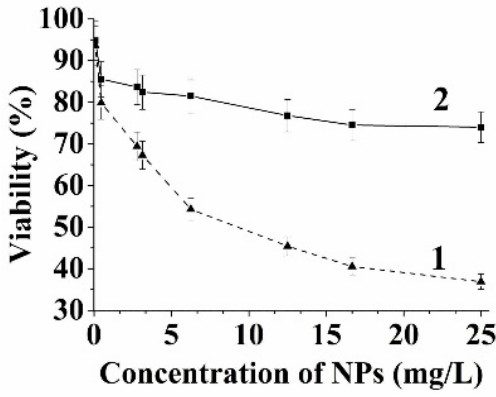

**Figure 11.** Cell viability of test culture of *Chlorella vulgaris* Beijer in the environment of $CoFe_2O_4$ nanoparticles (1) before and (2) after one gold deposition cycle.

The sulfur-containing amino acid L-methionine plays an important role in metabolism, methylation and transmethylation reactions in organisms [57–59], as well as in metal bonding in the structure of many enzymes [60,61]. It is known that methionine is one of the few reducing amino acids. Moreover, as shown in [35,42], methionine does not form separate gold crystals in the solution bulk, reducing unwanted side reactions and gold waste. We used methionine as both a biocompatible reducing agent and an anchor between $CoFe_2O_4$ and gold.

While many studies have been conducted on the topic, the mechanism of the methionine oxidation by $[AuCl_4]^-$ ions is not completely clear. According to [47,62–64], the reaction occurs in two steps. At the first step, methionine rapidly coordinates as a bidentate S- and N-donor ligand with Au (III) ion and forms an intermediate complex $[AuCl_2(HMet)]^{2+}$ (for acidic solutions) or $[AuCl_2(Met)]^+$ (for alkaline solutions). At the second step, the intermediate reacts with the second methionine molecule to form methionine sulfoxide as a product of methionine oxidation and the hydrolysis of the inter-

mediate compound. The $[AuCl_2]^-$ formed during the process then disproportionates, forming $[AuCl_4]^-$ and metallic gold. However, other oxidation routes are also possible. In Scuderi's research [65] and Asmus's work [66,67], the methionine oxidation process is complex and influenced by many conditions, and sulfoxide is not the sole product as it has been shown earlier. Scuderi found ammonia and $CO_2$ among the oxidation products of methionine, which indicates the decarboxylation and deamination of the amino acid. Additionally, the dimerization of methionine (or its decomposition product) occurs during the oxidation process with the formation of a two-center three-electron S-S bond (2c-3e), as noted by many researchers [50,68,69]. According to the XPS data, we observed this product during the deposition of gold on the surface of the nanoparticles. Moreover, its concentration increases with the increase in the number of gold deposition cycles. This finding has been obtained earlier in our research [42].

As discussed above, in this paper, 10–15 nm magnetic nanoparticles of cobalt ferrite, decorated with 2 nm gold nanoclusters, were obtained. It is worth noting that the nano-materials decorated with gold are of interest for catalytic applications [4,5,20–26,70,71]. The gold-decorated nanoparticles were synthesized by Félix et al. [22], using polyeth-yleneimine-treated $Fe_3O_4$ as magnetic cores and sodium borohydride as a reducing agent; they were also synthesized by Silvestri et al. [5], using $CoFe_2O_4$ and 2,3-meso-dimercapto succinic acid as a bifunctional organic ligand. In [72], in order to obtain core-shell nano-particles, formaldehyde was used as an initiator of gold deposition on the surface of sil-icon dioxide. Goon et al. [73] grew gold shell on $Fe_3O_4$—polyethyleneimine–Au seed by the iterative reduction of $HAuCl_4$ with $NH_2OH \cdot HCl$.

Despite the absence of a solid gold shell, the toxicity of the hybrid $CoFe_2O_4/Au$ na-noparticles was greatly reduced, even after one cycle of gold deposition. The obtained nanomaterial shows ferrimagnetic behavior. When ferrimagnetic nanoparticles are placed in external alternating magnetic fields, they can rapidly be heated due to the ab-sorption of the electromagnetic field energy [74]. We believe this feature, coupled with low toxicity, makes $CoFe_2O_4/Au$ nanoparticles suitable candidates for potential applica-tions in magnetothermal therapy. Additionally, such particles can be easily directed to certain organs and tissues, together with drug molecules attached to their surface, with the help of an external magnetic field.

## 4. Conclusions

In this paper, the synthesis of magnetic $CoFe_2O_4/Au$ HNPs from cobalt ferrite and $H[AuCl_4]$ using L-methionine was demonstrated. Cobalt ferrite nanoparticles with mean size of 11,8 ± 2,3 nm as magnetic cores were prepared by a simple, fast and easily re-producible method, where the strong-base anion-exchange resin in OH form was used as a precipitant of cobalt and ferric ions. To attach gold to the $CoFe_2O_4$ surface, the direct reduction of Au(III) by methionine, which also acts as an anchor between the nanoparti-cle surface and gold, was performed. The gold deposition was repeated up to 10 times under the same conditions. In contrast to our previous study [42], a continuous gold shell on the surface of the $CoFe_2O_4$ nanoparticles was not formed, even after 10 cycles of gold deposition. The obtained results showed only a minor increase in the size of gold nanoclusters: from 2.12 ± 0.15 nm to 2.46 ± 0.13 nm, with the number of the nanoclusters remaining unchanged during multiple repeats. Furthermore, methionine reduces gold only on the surface of cobalt ferrite nanoparticles, not in the solution. We speculate that future studies should include using an additional reducing agent, stronger than methio-nine, to form the gold shell on the magnetic core surface. The TEM images of the HNPs, obtained after 1, 6 and 10 gold deposition cycles, showed that a continuous gold shell on the surface was not formed, and the surface of cobalt ferrite was covered with gold nanoclusters, which slightly increased with an increase in the number of gold deposition cycles without any change in surface density. Therefore, we assume that an additional reducing agent, stronger than methionine, should be used to form the gold shell on the magnetic core surface.

In accordance with the magnetometry data, $CoFe_2O_4/Au$ nanoparticles demonstrate ferrimagnetic behavior ($Ms = 27.04$ emu/g, $Hc = 600$ Oe at 300 K) corresponding to that for bare cobalt ferrite nanoparticles. Moreover, a decrease in the magnetic saturation and residual magnetization values per mass of the sample was observed for $CoFe_2O_4/Au$ due to the presence of gold and organic substances (amino acid methionine and its oxidation products). The comparative study of the toxicity of the synthesized HNPs on a test organism of the unicellular green alga, *Chlorella vulgaris*, showed that the toxic effect of $CoFe_2O_4/Au$ nanoparticles obtained was noticeably reduced, even after one gold deposition cycle, compared with that of uncoated cobalt ferrite nanoparticles. The low toxicity and suitable magnetic properties of $CoFe_2O_4/Au$ nanoparticles make them potential candidates for applications in magnetothermal therapy. In addition, the $CoFe_2O_4/Au$ system is of interest for catalytic applications, because it combines the cobalt ferrite magnetic properties with the catalytic features of Au clusters. Such a hybrid catalyst material has the higher efficiency and the lower cost due to the larger Au surface area, compared with gold NPs. Additionally, it can be easily and rapidly separated from the products and reaction medium, using an external magnetic field, and reused in batch reactions.

**Supplementary Materials:** The following are available online at www.mdpi.com/2075-4701/11/5/705/s1, **Figure S1**: TGA curve (black) and DSC curve (blue) of $CoFe_2O_4/Au$ NPs after 10 gold deposition cycles. **Figure S2:** Optical absorption spectra of gases ($H_2O$ is not determined) evolved during TGA (thermal decomposition of organic part of $CoFe_2O_4/Au$ NPs after 10 gold deposition cycles).

**Author Contributions:** Conceptualization, S.S., A.P., T.T. and D.K.; formal analysis, A.S., S.S., A.P., T.T., D.K and Y.M.; investigation, A.P., T.T., Y.M., D.K., A.A., Y.G., M.V., A.S., S.Z. and D.V.; methodology, S.S.; supervision, S.S.; validation, S.S.; writing—original draft, S.S., A.P. and T.T.; writing—review and editing, Y.M. All authors have read and agreed to the published version of the manuscript.

**Funding:** This research received no external funding.

**Institutional Review Board Statement:** Not applicable.

**Informed Consent Statement:** Not applicable.

**Data Availability Statement:** No significant data in this study were created or analyzed, aside from the data presented here.

**Acknowledgments:** The authors thank the Federal Research Center, "Krasnoyarsk Science Center of the Siberian Branch of the Russian Academy of Sciences", for using its facilities.

**Conflicts of Interest:** The authors declare no conflict of interest.

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
