# Peer review of "Hybrid Nanoparticles Based on Cobalt Ferrite and Gold: Preparation and Characterization"

_metals, doi:10.3390/met11050705_

Round 1
Reviewer 1 Report
Dear authors,
The work entilted “Hybrid nanoparticles based on cobalt ferrite and gold: preparation and characterization” presents a fabrication route for the formation of Co ferrite nanoparticles with Au clusters on their surface. The nanoparticles were studied by different experimental techniques.
The introduction section provides a good overview of the question the authors try to tackle, including recent literature. However, the authors failed to clearly explain the novelty this work compared to the vast literature of ferrites covered with Au.
The Materials & Methods section is concise and correct.
In the results section, as a general comment, there are questions that should be better explained/clarified, as sometimes the description of the results obtained seems contradictory.
The decimals are sometimes written with commas. Please, check and change them for points.
Page 4 line 185. The authors state “the precipitate obtained is free from impurity…”. However, this statement is difficult to accept if we take into account that chemical routes inevitably lead to the presence of organic compounds on the surface. If the authors try to mention some particular feature (i.e.: concerning the crystalline structure). Please, clarify.
Page 4 line 194. The authors mentioned a narrow size distribution and provide an interval of average size distribution of 10-20 nm, which implies a variation in size of 100 % within this interval. I imagine that, if you have this agglomeration, it is difficult to provide a more accurate value. However, looking at the scale bar, it seems to me that most of the nanoparticles present an average size clearly below 20 nm.
In the x values of the histogram of fig. 1c, the first value corresponds to an interval of 5 nm, while the other corresponds to 10 nm. Please try to unify. If you are able to provide smaller intervals, it would be better to give more accurate information about the size distribution.
Page 5, line 209. When the authors mentioned “The L-methionine and H[AuCl4] treatments of the nanoparticles are numerously repeated.” What do they try to mean? It is not clear to me if they try to mention the fact that, by repeating this process, they obtain numerous Au0 seeds deposits on the ferrite or they try to explain any other thing. Please clarify. Reading the rest of the manuscript I assume that they refer to the 1, 6 and 10 deposition cycles. If so, provide the values instead using “numerously repeated.”
The authors also mentioned an initial treatment with L-methionine prior to the HAuCl4 addition. For a person that is not familiar with the process, the importance of this step is not clear. However, afterwards the authors made a comparison in figure 2 before and after this step. Furthermore, in line 226 the authors mentioned “Nevertheless, it was observed only insignificant sorption of the methionine molecules on NPs, therefore, the nature of the bonding between amino acid and CoFe2O4 requires additional research.” A whole paragraph describing the FTIR to conclude that “requires additional research”. The authors must explain better all this part, the aim followed in the step where L-methionine is added and if it is reached or not. Reading the manuscript is not clear.
Figure 2: check label of Y axis. Change “Absorbtion” for “Absorption”.
Where does the Al comes from in Figure 4?
Line 247: Please, revise the sentence “The obtained after 1 gold deposition cycle particles”. Do you mean the obtained particles after 1 gold deposition cycle?. Please check.
Figure 5: Are the authors sure that is correct to provide the extinction in a.u.?
In the XPS section, the authors mentioned “The presence of Fe 2p and Co 2p in all spectra indicates that a solid gold shell was not formed”. Considering the size of the Au clusters, even though they ended in a continuous film, Co and Fe should be detected. Therefore, these results are not an evidence of the absence of a continuous Au layer.
In addition, in line 268 the authors mentioned that “the partial desorption of adsorbed gold nanoparticles is caused by additional steps of the sonication treatment, magnetic separation and washing of CoFe2O4/Au NPs during extra gold deposition cycles”. However, when explaining the TEM images, they mentioned that “the density of the gold nanoclusters on the cobalt ferrite surface does not change with the increase in the number of gold deposition cycles.” In my opinion both facts are contradictory. Please clarify.
Another question that grabs my attention is that the authors claim to have CoFe2O4, although the proportion of Co and Fe in the three samples are far from the expected. In all of them Co is more abundant than Fe. Do the authors have any explanation about this point? In figure 1 they presented the XRD of the CoFe2O4 particles before the process in which Au is added. It would be interesting to have this information as the authors have already done in previous similar works (i.e. [38]).
If the calibration was performed at 284.45 eV and the main C peak is at 284.6 eV according to table 1, what is the main C specie? Please revise that all values are correct.
For a better comparison between the samples, in my opinion is better to compare the XPS spectra stacked in vertical. This is especially relevant for the Au 4f (figure 8), where small changes occur. Additionally, if they put vertical lines to indicate the components, it also difficult the observance of shoulders in the spectra (i.e.: blue line in fig 8c).
When the authors described the components in the text, they put the binding energy corresponding to one of the spectra, but the BE used for the other are not exactly the same. For instance, the component Au+ at 84.9 eV, in figure 8c is at 85.1 eV. You can refer in the text to this component as 85.0 ± 0.1 eV and label it in the graph as Au+. This will help the reader. In addition, use the same colour code for the same components. The component in red of Fig. 8a is blue in 8b & 8c and vice versa.
I don’t understand the values provided in Table 2. The sum of the components does not reach neither 100 % nor the total amount of Au of each sample. The authors must explain better the data provided. I think that the most interesting analysis is the % of each component referred to 100 % Au.
Concerning the magnetic characterization, how the authors calculate the amount of material they have in the system to provide the value in emu/g?
Why do the authors display only the magnetic and cell viability results of the sample with one cycle? I can understand that, if Au depletion occurs, it is the most interesting for their purposes and maybe they did not measure the response of the other samples. Please, specify and provide the reader the information about your choice.
In Line 343 the authors claim “An insignificant decrease in the magnetic saturation and residual magnetisation 343 values…”. However, with the small values registered, the decrease observed after Au addition is close to 20 %. In my opinion, a decrease of 20 % is not insignificant. Please revise.
Line 348: The authors mentioned “The magnetic properties strongly depend on the prehistory of the sample (annealing, preparation method, size and morphology of crystallites)”. In my opinion, the size of the particles is not prehistory, as the authors named it. It is a consequence of the fabrication process. Furthermore, it is one of the key factors that determine the Ms values, as for instance Rajendran et al. [Journal of Magnetism and Magnetic Materials 232 (2001) 71–83] reported for the same fabrication procedure.
Figure 10: check the figure caption (subscript CoFe2O4)
How can the authors be sure that the better toxicity of the samples is due to the small amount of Au on the NPs (5% on the surface in the best case) and not due to the great number of organic compounds around the nanoparticles? If the nanoparticles have amino acids on their surface, one tends to think that this system is more biocompatible.
Line 392: The authors mentioned “As discussed above, in this paper, 10–15-nm magnetic nanoparticles of cobalt ferrite decorated with 2-nm gold nanoclusters have been obtained.” However, in figure 1, the histogram presents the maximum in the 10-20 nm range as the authors mentioned in line 194. Please revise. Keep in mind that I previously suggest to uniform the x axis of figure 1. Maybe if you put intervals of 5 nm, this statement is correct, but not with the information currently provided.
When the authors made a summary just before the conclusions, they mentioned about Au clusters (line 402): “… the number of the nanoclusters remaining unchanged during multiple repeats”. However, when they presented the XPS, the proportion of Au decreases from 5 to 3%. In line 267 they attributed this decrease to the repetition of the cycles: “The concentration of gold on the NPs surface decreases with an increase in the number of deposition cycles. We suppose that the partial desorption of adsorbed gold nanoparticles is caused by additional steps of the sonication treatment, magnetic separation and washing of CoFe2O4/Au NPs during extra gold deposition cycles.” The authors must clarify this point. As in the XPS the Co and Fe content increases, I suppose that the decrease in the Au % is not related to a larger proportion of organic compounds on the surface derived from the fabrication process.
The conclusions section is only second summary of the results as they did in the three paragraphs previous to the conclusion section. In my opinion, the authors can shorten both, the summary previous to the conclusions as well as the conclusion section. The conclusions should avoid sentences like …the samples were measured by XPS if no XPS are commented afterwards. The authors should make the efort to write a real conclusion section and not just a summary of the work.
It seems that the method chosen by the authors to generate an Au shell is very aggressive and generate a progressive depletion. I can understand that, even if they don’t reach a complete capping layer as they did in other systems, it is interesting to have Au clusters on the surface. However, at the end, the magnetic response of the system is worse than the one without Au. The authors must provide a more convincing narrative of why the system is still interesting.
After all this comments, I can only recommend this manuscript for publication in Metals after major revisions.
Author Response
Reply: We are grateful to the reviewer for a thorough review and for posing many important questions. We have tried to respond to each question/statement and have also edited the revised manuscript. Please find below a detailed point-by-point response to all comments (reviewers’ comments in black, our replies in blue).
- The introduction section provides a good overview of the question the authors try to tackle, including recent literature. However, the authors failed to clearly explain the novelty this work compared to the vast literature of ferrites covered with Au.
Reply: We thank the reviewer for spending his precious time to review this article. While a wide number of papers relate to the production of hybrid nanoparticles, they examine the synthesis of materials mainly based on magnetite and gold. In comparison with magnetite, cobalt ferrite nanoparticles have better magnetic properties. The research on the synthesis of hybrid nanoparticles with the structure "cobalt ferrite - gold shell" have been conducted, however the main problems of number papers are the using non-biocompatible and environmentally unsafe reducing agents or/and the lack of reliable evidence for the production a dense gold shell on the surface of the magnetic core. A number of papers show synthesis of the magnetic cores decorated by gold seeds instead. The aim of the present research was to test the hypothesis about the possibility the further growth of the Au clusters to a solid shell is caused by methionine multi-stage gold reduction. Besides, we mean to examine what changes in the state of the system are produced with an increase in the gold deposition stages. To the best of our knowledge, ours is the first article examing the effect of multistage gold deposition on the CoFe2O4 nanoparticles surface coverage. In addition we provide a new efficient synthesis route for the production of the CoFe2O4 magnetic cores. The relevant text have been added in the manuscript and highlighted as well.
- The decimals are sometimes written with commas. Please, check and change them for points.
Reply: Thank you, we have corrected that.
- Page 4 line 185. The authors state “the precipitate obtained is free from impurity…”. However, this statement is difficult to accept if we take into account that chemical routes inevitably lead to the presence of organic compounds on the surface. If the authors try to mention some particular feature (i.e.: concerning the crystalline structure). Please, clarify. –
Reply: We use a strong-base anion-exchange resin in OH-form as a precipitant of cobalt and ferric ions. During the process, the anions of the solution (Cl-) would exchange with OH-ions of the resin. It means that Co2+, Fe3+ and OH– given off by the resin would combine to form hydroxides. It allows obtaining homogeneous nanoparticles without impurity ions (chloride ions of the CoCl2 and FeCl3). It may contain adsorbed organic substances (dextran or its derivatives), but the product obtained after calcination (CoFe2O4 nanoparticles) does not contain them and is a monophase cobalt ferrite. This is confirmed by the data of XRD and IR-Fourier spectroscopy. The absence of the peaks at 1000–1300 cm-1 and 2000–3000 cm-1 in the sample annealed at 600 ˚С, confirmed the nonexistence of the O–H mode, C–O mode, and C=H stretching-mode of organic sources. FT-IR analysis is very useful for establishing the presence of unwanted ions and molecules, which may pollute the crystal lattice during preparation.
- Page 4 line 194. The authors mentioned a narrow size distribution and provide an interval of average size distribution of 10-20 nm, which implies a variation in size of 100 % within this interval. I imagine that, if you have this agglomeration, it is difficult to provide a more accurate value. However, looking at the scale bar, it seems to me that most of the nanoparticles present an average size clearly below 20 nm.
In the x values of the histogram of fig. 1c, the first value corresponds to an interval of 5 nm, while the other corresponds to 10 nm. Please try to unify. If you are able to provide smaller intervals, it would be better to give more accurate information about the size distribution.
Reply: Thank you very much for your comment. The comments are greatly appreciated and taken into account. The figure has been revised.
- Page 5, line 209. When the authors mentioned “The L-methionine and H[AuCl4] treatments of the nanoparticles are numerously repeated.” What do they try to mean? It is not clear to me if they try to mention the fact that, by repeating this process, they obtain numerous Au0 seeds deposits on the ferrite or they try to explain any other thing. Please clarify. Reading the rest of the manuscript I assume that they refer to the 1, 6 and 10 deposition cycles. If so, provide the values instead using “numerously repeated.”
Reply: We are thankful for your suggestion. The corrected version of the text is «The L-methionine and H[AuCl4] treatments of the nanoparticles were carried out in 1, 6 and 10 deposition cycles»
- The authors also mentioned an initial treatment with L-methionine prior to the HAuCl4 For a person that is not familiar with the process, the importance of this step is not clear. However, afterwards the authors made a comparison in figure 2 before and after this step. Furthermore, in line 226 the authors mentioned “Nevertheless, it was observed only insignificant sorption of the methionine molecules on NPs, therefore, the nature of the bonding between amino acid and CoFe2O4 requires additional research.” A whole paragraph describing the FTIR to conclude that “requires additional research”. The authors must explain better all this part, the aim followed in the step where L-methionine is added and if it is reached or not. Reading the manuscript is not clear.
Reply: Thank you very much for your comment. The synthesis was carried out in accordance with the method proposed in [1]. At first, cobalt ferrite nanoparticles were dispersed in L-methionine (20 mL, 0.03 mol/L) by ultrasonic agitation for 30 min. Than, chlorauric acid and NaOH were added to the mixture. No additional methionine treatment was performed. It is supposed [1] that upon dispersion of ferrite NPs in the methionine solution amino acid interacts with the surface of cobalt ferrite NPs through the carboxyl bond. We use FTIR spectroscopy to determine the presence of methionine on the surface of magnetic nuclei and the nature of the bonds between the amino acid and CoFe2O4 nanoparticles surface. Due to the low intensity of the spectrum in this area, it is difficult to draw unambiguous conclusions.
- Figure 2: check label of Y axis. Change “Absorbtion” for “Absorption”.
Reply: Thank you, we have corrected that.
- Where does the Al comes from in Figure 4?
Reply: The source of the Al and Cu signals is the sample holder for EDX analysis
- Line 247: Please, revise the sentence “The obtained after 1 gold deposition cycle particles”. Do you mean the obtained particles after 1 gold deposition cycle?. Please check.
Reply: Thank you, we have corrected that.
- Figure 5: Are the authors sure that is correct to provide the extinction in a.u.?
Reply: Thank you, we have corrected that.
- In the XPS section, the authors mentioned “The presence of Fe 2p and Co 2p in all spectra indicates that a solid gold shell was not formed”. Considering the size of the Au clusters, even though they ended in a continuous film, Co and Fe should be detected. Therefore, these results are not an evidence of the absence of a continuous Au layer.
Reply: We are grateful for your valuable comment. According to the TEM data, the dense gold shell on the surface of the magnetic core had not formed. We had corrected the text.
- In addition, in line 268 the authors mentioned that “the partial desorption of adsorbed gold nanoparticles is caused by additional steps of the sonication treatment, magnetic separation and washing of CoFe2O4/Au NPs during extra gold deposition cycles”. However, when explaining the TEM images, they mentioned that “the density of the gold nanoclusters on the cobalt ferrite surface does not change with the increase in the number of gold deposition cycles.” In my opinion both facts are contradictory. Please clarify.
Reply: We agree with the reviewer on this important point. The density of gold nanoparticles on the surface of cobalt ferrite nanoparticles after 1, 6 and 10 deposition cycles was recalculated. It had founded that it decreases with the increase in the number of gold deposition cycles: 21.4 ± 2.7 Au clusters after one cycle, 19.0 ± 1.2 after six cycles and 18.0 ± 1.4 after ten cycles. In addition, as can be seen from the XPS data (Table 2), the decrease in the amount of Au on the surface may be due to the removal of Au+ ions during washing. We have revised the text and highlighted as well.
- Another question that grabs my attention is that the authors claim to have CoFe2O4, although the proportion of Co and Fe in the three samples are far from the expected. In all of them Co is more abundant than Fe. Do the authors have any explanation about this point? In figure 1 they presented the XRD of the CoFe2O4 particles before the process in which Au is added. It would be interesting to have this information as the authors have already done in previous similar works (i.e. [38]).
Reply: It is known that, the XPS determines the concentration of metals on the surface, and not the total amount of metals in the volume of the phase. According to the XRD data, we obtained a monophase stoichiometric product - CoFe2O4 (Co-25.1 wt %, Fe-47.6 wt %, O-27.2 wt %)
- If the calibration was performed at 284.45 eV and the main C peak is at 284.6 eV according to table 1, what is the main C specie? Please revise that all values are correct.
Reply: The C 1s bands actually consist of several components, including a contribution of the HOPG support. To avoid confusion, we removed the column with binding energies in Table 1, thank you for the remark.
- For a better comparison between the samples, in my opinion is better to compare the XPS spectra stacked in vertical. This is especially relevant for the Au 4f (figure 8), where small changes occur. Additionally, if they put vertical lines to indicate the components, it also difficult the observance of shoulders in the spectra (i.e.: blue line in fig 8c).
Reply: We have changed the arrangement of the figures.
- When the authors described the components in the text, they put the binding energy corresponding to one of the spectra, but the BE used for the other are not exactly the same. For instance, the component Au+at 84.9 eV, in figure 8c is at 85.1 eV. You can refer in the text to this component as 85.0 ± 0.1 eV and label it in the graph as Au+. This will help the reader. In addition, use the same colour code for the same components. The component in red of Fig. 8a is blue in 8b & 8c and vice versa.
Reply: We are thankful for your suggestion, we have corrected that.
- I don’t understand the values provided in Table 2. The sum of the components does not reach neither 100 % nor the total amount of Au of each sample. The authors must explain better the data provided. I think that the most interesting analysis is the % of each component referred to 100 % Au.
Reply: Thank you, we have corrected that.
- Concerning the magnetic characterization, how the authors calculate the amount of material they have in the system to provide the value in emu/g?
Reply: The magnetic parameters values were calculated according to the sample weights used in the SQUID measurements
- Why do the authors display only the magnetic and cell viability results of the sample with one cycle? I can understand that, if Au depletion occurs, it is the most interesting for their purposes and maybe they did not measure the response of the other samples. Please, specify and provide the reader the information about your choice.
Reply: Thank you very much for your comment. According to TEM and ХPS data, the density of gold clusters on the surface of cobalt ferrite nanoparticles does not increase during the increase in the number of gold deposition cycles, moreover Au depletion is observed. We use the sample with the highest gold nanoparticles concentration.
- In Line 343 the authors claim “An insignificant decrease in the magnetic saturation and residual magnetisation 343 values…”. However, with the small values registered, the decrease observed after Au addition is close to 20 %. In my opinion, a decrease of 20 % is not insignificant. Please revise.
Reply: Thank you, we have corrected that. Corrected version of the text: «A decrease in the magnetic saturation and residual magnetisation values per mass of the sample was observed for CoFe2O4/Au due to the presence of diamagnetic gold».
- Line 348: The authors mentioned “The magnetic properties strongly depend on the prehistory of the sample (annealing, preparation method, size and morphology of crystallites)”. In my opinion, the size of the particles is not prehistory, as the authors named it. It is a consequence of the fabrication process. Furthermore, it is one of the key factors that determine the Ms values, as for instance Rajendran et al. [Journal of Magnetism and Magnetic Materials 232 (2001) 71–83] reported for the same fabrication procedure.
Reply: Thank you, we have corrected that. Corrected version of the text: «The magnetic properties strongly depend on the fabrication procedure of the sample (annealing, preparation method, size and morphology of crystallites) [47].»
- Figure 10: check the figure caption (subscript CoFe2O4)
Reply: Thank you, we have corrected that.
- How can the authors be sure that the better toxicity of the samples is due to the small amount of Au on the NPs (5% on the surface in the best case) and not due to the great number of organic compounds around the nanoparticles? If the nanoparticles have amino acids on their surface, one tends to think that this system is more biocompatible.
Reply: We are thankful for your suggestion and have added this idea to the manuscript and highlighted the text.
- Line 392: The authors mentioned “As discussed above, in this paper, 10–15-nm magnetic nanoparticles of cobalt ferrite decorated with 2-nm gold nanoclusters have been obtained.” However, in figure 1, the histogram presents the maximum in the 10-20 nm range as the authors mentioned in line 194. Please revise. Keep in mind that I previously suggest to uniform the x axis of figure 1. Maybe if you put intervals of 5 nm, this statement is correct, but not with the information currently provided.
Reply: Thank you, we have revised figure1b.
- When the authors made a summary just before the conclusions, they mentioned about Au clusters (line 402): “… the number of the nanoclusters remaining unchanged during multiple repeats”. However, when they presented the XPS, the proportion of Au decreases from 5 to 3%. In line 267 they attributed this decrease to the repetition of the cycles: “The concentration of gold on the NPs surface decreases with an increase in the number of deposition cycles. We suppose that the partial desorption of adsorbed gold nanoparticles is caused by additional steps of the sonication treatment, magnetic separation and washing of CoFe2O4/Au NPs during extra gold deposition cycles.” The authors must clarify this point. As in the XPS the Co and Fe content increases, I suppose that the decrease in the Au % is not related to a larger proportion of organic compounds on the surface derived from the fabrication process.
Reply: Thank you very much for your comment. The density of gold nanoparticles on the surface of cobalt ferrite nanoparticles after 1, 6 and 10 deposition cycles was recalculated. It had founded that it decreases with the increase in the number of gold deposition cycles: 21.4 ± 2.7 Au clusters after one cycle, 19.0 ± 1.2 after six cycles and 18.0 ± 1.4 after ten cycles. In addition, as can be seen from the XPS data (Table 2), the decrease in the amount of Au on the surface may be due to the removal of Au+ ions during extra washing. We have revised the text and highlighted as well.
- The conclusions section is only second summary of the results as they did in the three paragraphs previous to the conclusion section. In my opinion, the authors can shorten both, the summary previous to the conclusions as well as the conclusion section. The conclusions should avoid sentences like …the samples were measured by XPSif no XPS are commented afterwards. The authors should make the efort to write a real conclusion section and not just a summary of the work.
Reply: These sections have been revised.
- It seems that the method chosen by the authors to generate an Au shell is very aggressive and generate a progressive depletion. I can understand that, even if they don’t reach a complete capping layer as they did in other systems, it is interesting to have Au clusters on the surface. However, at the end, the magnetic response of the system is worse than the one without Au. The authors must provide a more convincing narrative of why the system is still interesting.
Reply: Biomedical applications typically require soft magnetic materials. Besides, gold nanoparticles extensively used as efficient redox catalyst materials. CoFe2O4/Au systems are of interest for catalytic applications, combining recovery and reuse, as well as increased activity under illumination [2-3]. Such hybrid catalyst material has the higher efficiency and the lower cost due to the larger Au surface area comparing gold NPs. It is also worth noting the possibility of separating the catalyst using an external magnetic field.
[1] Mikalauskaite, A.; Kondrotas, R.; Niaura, G.; Jagminas, A. Gold-coated cobalt ferrite nanoparticles via methionine-induced reduction. J. Phys. Chem. C 2015, 119, 17398-17407.
[2] Wang, H.; Shen, J.; Li, Y.; Wei, Z.; Cao, G; Hong, K.; Banerjee, P.; Zhou, S. Porous carbon protected magnetite and silver hy-brid nanoparticles: morphological control, recyclable catalysts, and multicolor cell imaging. ACS Appl. Mater. Interfaces 2013, 5, 9446-9453.
[3] Saire-Saire, S.; Barbosa, E.C.M.; Garcia, D.; Andrade, L.H.; Garsia-Segura, S.; Camargo, P.H.C.; Alarcon, H. Green synthesis of Au decorated CoFe2O4 nanoparticles for catalytic reduction of 4-nitrophenol and dimethylphenylsilane oxidation. RSC Adv. 2019, 9, 22116-22123].

Reviewer 2 Report
Saikova et al. presented the fabrication of decorated CoFe2O4 nanoparticles with 2nm Au nanoparticles. They presented an easy method and performed a full characterization. After a careful reading of the manuscript, and despite the fact that plenty of different routes can fabricate CoFe2O4 nanoparticles, and then use the same methodology here presented for decorating them, I found that the membrane synthetic approach can be further exploited for obtaining nanoparticles without byproducts. However, authors described a method to fabricate CoFe2O4 nanoparticles with high reproducibility, but no statistical results are shown. Some experiments are missing and some corrections of the data obtained should be performed. Moreover, I think authors should extend the target audience beyond the biological applications: the methodology developed could be interesting for catalysis but also for area selective depositions. Then, to improve the quality of the manuscript, authors should take into account the following recommendations before publication:
1. Abstract: what is a deposition cycle? It has not been mentioned before.
2. Introduction: As in at the end of the manuscript, references towards applications on the catalysis field should be included, since I consider it also as a target field. The same for area selective deposition, where Au wants to be deposited only on Au surfaces but not on CoFe2O4, preventing the generation of new nuclei on the magnetic surface.
3. Results: If authors claim that the method is reproducible, they must show enough statistical evidence that the same particle size is obtained. Statistical information is missing along all the research.
4. Fig. 1 Histogram should have higher size resolution for observing better the obtained particle distribution. At least 2-3 nm steps
5. Statistical information about the reproducibility must be included. From the synthetic but also characterization point of view.
6. Fig. 2. I would recommend including the FTIR spectra as transmittance (to be consistent with the literature). Peaks identification on the plot is required. Why both spectra a and b have different X range? It should be from 500 to 4000 cm-1
7. Fig. 3. Compared with the initial histogram and TEM images, the CoFe2O4 nanoparticles seems to be larger after Au deposition. Why is that? Why the particles seem to be aggregated?
8. What is the effect of the pH when preparing the hybrid nanoparticles? Specially if you vary it from 10 to higher and lower pH.
9. Based on the UV-Vis spectra, could you approximate the separation yield of the hybrid nanoparticles from the solution?
10. SQUID. Have been the SQUID measurements in emu/g corrected against the amount of organic material surrounding the nanoparticles? Using TGA, authors can calculate the weight % of organic part and then correct the magnetization per mass unit, which is due only to the CoFe2O4 NPs and not the organic material. (See the XPS data, where authors proved the high amount of organic material in their samples). Then, the magnetization would be much more approximate to the expected values (which have been reported in function of the amount of inorganic material without any capping ligand or organic material in the media). I consider that this would explain better why the real values are lower than expected, in addition to the surface disorder explanation included and the possibility of having amorphous structures. Authors should provide the % in weight of organic material of their samples via TGA values and correct the SQUID measurements using the value unless they did it in another way, and then they should provide much more specific data in the manuscript.
11. Authors claim that the hybrid nanoparticles can be used for magnetotherapy. However, and as authors proved, the cytotoxicity is quite high compared to pure covered nanoparticles. Authors should describe better why these particles are better that core-shell nanoparticles. In fact, I am much more convinced that these particles are useful for catalysis, as authors mentioned in one sentence, and I think this should be a better approach for future applications of the hybrid system. I consider that authors should also explain in the introduction the advantages for catalysis compared to the core shell and pure gold nanoparticles (magnetic particle separation, larger Au surface area…)
12. Authors should stress the selective deposition of Au on Au and not on the CoFeO4 when increasing the number of cycles. This is very interesting for different current technological fields where selective area deposition is required. This chemical method is also showing that it is possible to do it, since by increasing the number of cycles, more Au is deposited but only on Au NPs and not generating new nuclei (following authors’ results). I would suggest the authors to investigate further the selective deposition in future research works.
I encourage the authors to take into consideration the questions and suggestions to improve the overall work quality.
Author Response
Reply: We are grateful to the reviewer for a thorough review and for posing many important questions. We have tried to respond to each question/statement and have also edited the revised manuscript. Please find below a detailed point-by-point response to all comments (reviewers’ comments in black, our replies in blue).
Saikova et al. presented the fabrication of decorated CoFe2O4 nanoparticles with 2nm Au nanoparticles. They presented an easy method and performed a full characterization. After a careful reading of the manuscript, and despite the fact that plenty of different routes can fabricate CoFe2O4 nanoparticles, and then use the same methodology here presented for decorating them, I found that the membrane synthetic approach can be further exploited for obtaining nanoparticles without byproducts. However, authors described a method to fabricate CoFe2O4 nanoparticles with high reproducibility, but no statistical results are shown. Some experiments are missing and some corrections of the data obtained should be performed. Moreover, I think authors should extend the target audience beyond the biological applications: the methodology developed could be interesting for catalysis but also for area selective depositions. Then, to improve the quality of the manuscript, authors should take into account the following recommendations before publication:
1. Abstract: what is a deposition cycle? It has not been mentioned before.
Reply. We are thankful for your concern. 1, 6, and 10 are the number of gold deposition cycles. We have revised the abstract and highlighted the changed text.
- Introduction: As in at the end of the manuscript, references towards applications on the catalysis field should be included, since I consider it also as a target field. The same for area selective deposition, where Au wants to be deposited only on Au surfaces but not on CoFe2O4, preventing the generation of new nuclei on the magnetic surface.
Reply: We are thankful for your suggestion. The above-mentioned references [20-26] have been cited in the manuscript.
- Results: If authors claim that the method is reproducible, they must show enough statistical evidence that the same particle size is obtained. Statistical information is missing along all the research.
Reply: We thank the reviewer for spending his precious time to review this article. This study focuses on the synthesis of hybrid CoFe2O4/Au nanoparticles and the testing of the hypothesis about the possibility the further growth of the Au clusters to a solid shell is caused by methionine multi-stage gold reduction. Besides, we mean to examine what changes in the state of the system are produced with an increase in the gold deposition stages. However, we provide size distribution histogram of cobalt ferrite particles synthesised by anion-exchange resin co-precipitation (fig.1c), we calculate mean sizes Au clusters and their surface density. More statistical information is available in our previous papers devoted the anion exchange synthesis technique [1-3].
- Fig. 1 Histogram should have higher size resolution for observing better the obtained particle distribution. At least 2-3 nm steps
Reply: Thank you, we have revised figure1b.
- Statistical information about the reproducibility must be included. From the synthetic but also characterization point of view.
Reply: We provide size distribution histogram of cobalt ferrite particles synthesised by anion-exchange resin co-precipitation (fig.1c), we calculate mean sizes Au clusters and their surface density. The size of the gold nanoclusters slightly increases with an increase in the number of gold deposition cycles: 2.12 ± 0.15 nm for one cycle, 2.15 ± 0.15 nm for six cycles and 2.46 ± 0.13 nm for ten cycles. The density of the gold nanoclusters on the cobalt ferrite surface does insignificant decrease with the increase in the number of gold deposition cycles: 21.4 ± 2.7 Au NPs for one cycle, 19.0 ± 1.2 Au NPs for six cycles and 18.0 ± 1.4 Au NPs for ten cycles. More statistical information is available in our previous papers devoted the anion exchange synthesis technique [1-3].
- 2. I would recommend including the FTIR spectra as transmittance (to be consistent with the literature). Peaks identification on the plot is required. Why both spectra a and b have different X range? It should be from 500 to 4000 cm-1
Reply: We are thankful for your concern. Figure 2b shows an enlarged part of the spectrum from Figure 2a in the range from 1200 to 1800 cm-1, we have changed the caption to the figure for a better understanding. The spectra as Absorption are given quite often in the literature.
- Fig. 3. Compared with the initial histogram and TEM images, the CoFe2O4 nanoparticles seems to be larger after Au deposition. Why is that? Why the particles seem to be aggregated?
Reply: We agree with the reviewer on this important point. The mean size of the cobalt ferrite nanoparticles increase after Au deposition from 11.8±2.3 nm to 17.6±0.9nm for one cycle, 20.5±1.1 for six cycles, 21.6±1.2 nm for ten cycles. May be it is a result of particles aggregation during the reduction of gold on the surface of cobalt ferrite. Unfortunately, we don’t have a relevant answer to this question at the moment.
- What is the effect of the pH when preparing the hybrid nanoparticles? Specially if you vary it from 10 to higher and lower pH.
Reply: The pH value did not vary in this study and was 12,0.
- Based on the UV-Vis spectra, could you approximate the separation yield of the hybrid nanoparticles from the solution?
Reply: The supernatant solutions obtained after the synthesis were colourless, indicating the absence of gold nanoparticles in the solution. The small absorption at 580 nm is due to the presence of a small amount (near 5 %) of hybrid nanoparticles in the solution due to incomplete magnetic separation.
- SQUID. Have been the SQUID measurements in emu/g corrected against the amount of organic material surrounding the nanoparticles? Using TGA, authors can calculate the weight % of organic part and then correct the magnetization per mass unit, which is due only to the CoFe2O4 NPs and not the organic material. (See the XPS data, where authors proved the high amount of organic material in their samples). Then, the magnetization would be much more approximate to the expected values (which have been reported in function of the amount of inorganic material without any capping ligand or organic material in the media). I consider that this would explain better why the real values are lower than expected, in addition to the surface disorder explanation included and the possibility of having amorphous structures. Authors should provide the % in weight of organic material of their samples via TGA values and correct the SQUID measurements using the value unless they did it in another way, and then they should provide much more specific data in the manuscript.
Reply: We are grateful for your valuable comment. Unfortunately, in the present project, due to limited lab access caused by the COVID situations, we could not analyze the samples via TGA values and in the meanwhile, the samples were destroyed. We agree with the reviewer that weight % of the organic part can reduce the specific magnetic properties together with the gold on the surface. The corresponding remark was added to the manuscript.
- Authors claim that the hybrid nanoparticles can be used for magnetotherapy. However, and as authors proved, the cytotoxicity is quite high compared to pure covered nanoparticles. Authors should describe better why these particles are better that core-shell nanoparticles. In fact, I am much more convinced that these particles are useful for catalysis, as authors mentioned in one sentence, and I think this should be a better approach for future applications of the hybrid system. I consider that authors should also explain in the introduction the advantages for catalysis compared to the core shell and pure gold nanoparticles (magnetic particle separation, larger Au surface area…)
Reply: Thank you very much for your comment. The comments are greatly appreciated and taken into account. We have added the text to the manuscript and highlighted the changed text.
- Authors should stress the selective deposition of Au on Au and not on the CoFeO4 when increasing the number of cycles. This is very interesting for different current technological fields where selective area deposition is required. This chemical method is also showing that it is possible to do it, since by increasing the number of cycles, more Au is deposited but only on Au NPs and not generating new nuclei (following authors’ results). I would suggest the authors to investigate further the selective deposition in future research works.
Reply: We are thankful for your concern.
[1] Saikova, S. V., Trofimova, T. V., Pavlikov, A. Y., & Samoilo, A. S. (2020). Effect of Polysaccharide Additions on the Anion-Exchange Deposition of Cobalt Ferrite Nanoparticles. Russian Journal of Inorganic Chemistry, 65(3), 291–298. doi:10.1134/s0036023620030110
[2] Trofimova, T.V., Saikova, S.V., Panteleeva, M.V. et al. Anion-Exchange Synthesis of Copper Ferrite Powders. Glass Ceram 75, 74–79 (2018). https://doi.org/10.1007/s10717-018-0032-7]
[3] Ivantsov, R., Evsevskaya, N., Saikova, S., Linok, E., Yurkin, G., & Edelman, I. (2017). Synthesis and characterization of Dy 3 Fe 5 O 12 nanoparticles fabricated with the anion resin exchange precipitation method. Materials Science and Engineering: B, 226, 171–176. doi:10.1016/j.mseb.2017.09.016].

Reviewer 3 Report
In this work, hybrid nanoparticles based on cobalt ferrite and gold were prepared and characterized. The topic is appropriate for the journal and is relevant to current interest. The following comments should be addressed before publition:
1.The saturation magnetisation of nanoparticles before and after Au coating are relatively low, please explain the reason and propose new strategy to increase the magnetic parameters value.
2. How to control nanopartices size by varing the reaction parameters?
Author Response
Reply: We are grateful to the reviewer for a thorough review and for posing many important questions. We have tried to respond to each question/statement and have also edited the revised manuscript. Please find below a detailed point-by-point response to all comments (reviewers’ comments in black, our replies in blue).
1.The saturation magnetisation of nanoparticles before and after Au coating are relatively low, please explain the reason and propose new strategy to increase the magnetic parameters value.
Reply: We are thankful for your concern. The observed saturation magnetisation values (33.21 emu/g CoFe2O4 and 27.04 emu/g for CoFe2O4/Au) are noticeably lower than the bulk values (80.8 emu/g) [1]. It is known that decreasing the particle size decreases the saturation magnetisation of the material. It is generally attributed to increasing surface spin canting caused by the structural disorder of surface atoms [2]. Decreasing the particle size leads to an increase in the relative number of atoms in the surface layer. It is worth noting that these values were obtained at 298 K, and at lower temperatures they are greater. It is also known that cobalt ferrite nanoparticles having diameter < 10 nm are superparamagnetic at 300 K [3]. In addition, the organic part can reduce the specific magnetic properties as well as the gold on the surface.
- How to control nanopartices size by varing the reaction parameters?
Reply: The sizes of nanoparticles are highly dependent upon the synthesis route and conditions such as solute concentrations, its viscosity, pH of the solution, heating temperatures and time, pressure and presence/absence of the surface modifiers. Thus, by optimizing these parameters, the sizes of nanoparticles can be controlled. We considered this issue in detail relation to anion-exchange deposition of cobalt ferrite nanoparticles earlier [4].
[1]. Stein, C.R.; Bezerra, M.T.S.; Holanda, G.H.A.; André-Filho, J.; Morais, P.C. Structural and magnetic properties of cobalt fer-rite nanoparticles synthesized by co-precipitation at increasing temperatures. AIP Adv. 2018, 8, 056303.]
[2] Ammar, S.; Helfen, A.; Jouini, N.; Fiévet, F.; Rosenman, I.; Villain, F.; Molinie, P.; Danot, M. Magnetic properties of ultrafine cobalt ferrite particles synthesized by hydrolysis in a polyol medium. J. Mater. Chem. 2001, 11,186 -192.
[3] Coey, J.M.D. Magnetism and magnetic materials. Cambridge university press 2009.].
[4] Saikova, S. V., Trofimova, T. V., Pavlikov, A. Y., & Samoilo, A. S. (2020). Effect of Polysaccharide Additions on the Anion-Exchange Deposition of Cobalt Ferrite Nanoparticles. Russian Journal of Inorganic Chemistry, 65(3), 291–298. doi:10.1134/s0036023620030110]

Reviewer 4 Report
Review of “Hybrid nanoparticles based on cobalt ferrite and gold: preparation and characterization”
In this paper, the author provides a simple method to obtain cobalt ferrite HNPs. L-methionine was used as a reducing agent, an anchor between CoFe2O4 and gold and a stabilizing agent for nanoparticles. The gold deposition on the surface of CoFe2O4 cores with methionine was performed in several cycles. The properties of the materials were characterized by XRD, TEM and XPS, and the biomedical applications of the materials were explored. This manuscript is interesting, but I have the following suggestions as follows:
(1) I don't think the size of Au nanoclusters in TEM can be obtained accurately. So it's not reasonable to judge the increase of their size. Unless the author states a more specific way of judging, I think the size of gold particles is similar, uniform distribution is more appropriate.
(2) Can the author clearly explain why there is no a solid gold shell after ten cycles of gold deposition?
(3) The journal of Metals is most interested in manuscripts that clearly present new and novel ideas of interest to a general readership. I regret that the present manuscript does not meet the stringent requirements for acceptance and publication in Metals.
Considering the standards of the journal, the article needs to make the above modifications. Therefore, I recommended the rejection of the manuscript.
Author Response
Reply: We are grateful to the reviewer for a thorough review and for posing many important questions. We have tried to respond to each question/statement and have also edited the revised manuscript. Please find below a detailed point-by-point response to all comments (reviewers’ comments in black, our replies in blue).
In this paper, the author provides a simple method to obtain cobalt ferrite HNPs. L-methionine was used as a reducing agent, an anchor between CoFe2O4 and gold and a stabilizing agent for nanoparticles. The gold deposition on the surface of CoFe2O4 cores with methionine was performed in several cycles. The properties of the materials were characterized by XRD, TEM and XPS, and the biomedical applications of the materials were explored. This manuscript is interesting, but I have the following suggestions as follows:
- I don't think the size of Au nanoclusters in TEM can be obtained accurately. So it's not reasonable to judge the increase of their size. Unless the author states a more specific way of judging, I think the size of gold particles is similar, uniform distribution is more appropriate.
Reply: TEM is a direct method for obtaining information about the morphology and size of the resulting nanoparticles. There are a large number of works in which the size is determined according to the TEM data.
- Can the author clearly explain why there is no a solid gold shell after ten cycles of gold deposition?
Reply: A wide number of papers show synthesis of the magnetic cores decorated by gold seeds instead the obtaining of solid shell. The aim of the present research was to test the hypothesis about the possibility the further growth of the Au clusters to a solid shell is caused by methionine multi-stage gold reduction [1]. Besides, we mean to examine what changes in the state of the system are produced with an increase in the gold deposition stages. In addition, gold nanoparticles extensively used as efficient redox catalyst materials. CoFe2O4/Au systems are of interest for catalytic applications, combining recovery and reuse, as well as increased activity under illumination [2-3]. Such hybrid catalyst material has the higher efficiency and the lower cost due to the larger Au surface area comparing gold NPs. It is also worth noting the possibility of separating the catalyst using an external magnetic field.
- The journal of Metals is most interested in manuscripts that clearly present new and novel ideas of interest to a general readership. I regret that the present manuscript does not meet the stringent requirements for acceptance and publication in Metals.
Reply: We are thankful for your concern. We plan to publish this work in the Special Issue "Metal Containing Nanoparticles for Biomedical Applications" of journal «Metals».
[1] Mikalauskaite, A.; Kondrotas, R.; Niaura, G.; Jagminas, A. Gold-coated cobalt ferrite nanoparticles via methionine-induced reduction. J. Phys. Chem. C 2015, 119, 17398-17407.
[2] Wang, H.; Shen, J.; Li, Y.; Wei, Z.; Cao, G; Hong, K.; Banerjee, P.; Zhou, S. Porous carbon protected magnetite and silver hy-brid nanoparticles: morphological control, recyclable catalysts, and multicolor cell imaging. ACS Appl. Mater. Interfaces 2013, 5, 9446-9453.
[3] Saire-Saire, S.; Barbosa, E.C.M.; Garcia, D.; Andrade, L.H.; Garsia-Segura, S.; Camargo, P.H.C.; Alarcon, H. Green synthesis of Au decorated CoFe2O4 nanoparticles for catalytic reduction of 4-nitrophenol and dimethylphenylsilane oxidation. RSC Adv. 2019, 9, 22116-22123].

Round 2
Reviewer 1 Report
Dear authors,
Even though the manuscript has been improved, but there are still many inconsistencies in their manuscript that have not been solved. In addition, many of the comments that I made have an answer, but no reflection in the manuscript.
I consider that the manuscript is not suitable for publication in the present form as it still have unsolved questions.
Below you will find more detail explanation of these questions that will help you to improve the manuscript:
-The authors did not include any further explanation in the FTIR part as requested to help for a better understanding.
- Figure 4: If Al and Cu comes from the sample holder, please, put it in brackets in the figure caption.
- Concerning the Au desorption, I’m still not convinced about their explanation. The authors provided some numbers of a decrease in the number of Au clusters of TEM images. However, the decrease reported is within the experimental error and cannot explain the decrease in the Au content observed in the XPS analysis of above a 60 % from one sample to another. They mentioned that the decrease in the amount of Au on the surface may be due to the removal of Au+ ions during washing. Does it mean that this washing was not needed before the TEM analysis? The authors still need to clarify this point because is one of the major inconsistencies of the work.
- As regard the Co and Fe content, I agree with the authors that XPS provide information about the surface composition. In this point I can understand that they don’t have the exact proportion of a stoichiometric compound. However, in my opinion, this does not explain the variation in the Co/Fe ratio among the samples from 1.5 to 0.6 in this new version of the manuscript and between 1.75 and 0.6 in the former version.
In this point I realized that there was a part of my question missing. I wrote:
In figure 1 they presented the XRD of the CoFe2O4 particles before the process in which Au is added. It would be interesting to have this information as the authors have already done in previous similar works (i.e. [38]).
When I wanted to ask the following:
In figure 1 they presented the XRD of the CoFe2O4 particles before the process in which Au is added. It would be interesting to have this information AFTER Au ADDITION as the authors have already done in previous similar works (i.e. [38]). This will provide light to this point, to check that the CoFe2O4 nanoparticles are not affected by the treatment to add the Au clusters. Please add XRD of the samples with Au.
- Concerning the binding energy of the C peak, one assumes that if the authors use the BE HOPG is because is the main component of C 1s. It was just a question that I thought it could be a typographical error.
By the way, why do the authors removed not only the binding energies of table 1 but also all reference to the carbon, that is the main specie detected (close to 60%at in all samples) without any mention in the text?. These new values provide a non-realistic vision of the results obtained.
- In the core level spectra, the authors still put vertical lines to indicate the components in places where hinders the following of the shoulders of the original data. They must be careful with that. It makes more difficult the evaluation of the results for the reader.
- SQUID measurements: the authors did not include any reference in the text of why they only put the results of one sample (only in the answer of the question). Furthermore, they explained the decrease of Ms by the presence of diamagnetic Au. Au is diamagnetic in bulk. When you go to the nanoscale, you can find a vast literature discussing about induced ferromagnetism an other effects. What is clear is that Au is weakly polarizable and can contribute to the magnetic response of the system.
Here I emphasize the importance of XRD after Au addition in the system to understand the results obtained.
Author Response
Reply: We are grateful to the reviewer for a thorough review and have tried to respond to each question/statement and have revised the manuscript (reviewers’ comments in black, our replies in blue).
- The authors did not include any further explanation in the FTIR part as requested to help for a better understanding.
Reply: Thank you, we have changed the text for clarity.
- Figure 4: If Al and Cu comes from the sample holder, please, put it in brackets in the figure caption.
Reply: Thank you, we have corrected that
- Concerning the Au desorption, I’m still not convinced about their explanation. The authors provided some numbers of a decrease in the number of Au clusters of TEM images. However, the decrease reported is within the experimental error and cannot explain the decrease in the Au content observed in the XPS analysis of above a 60 % from one sample to another. They mentioned that the decrease in the amount of Au on the surface may be due to the removal of Au+ions during washing. Does it mean that this washing was not needed before the TEM analysis? The authors still need to clarify this point because is one of the major inconsistencies of the work.
Reply: According to the XPS data, after the first gold deposition cycle not only metallic Au fixes on the surface of the CoFe2O4 particles. The most of the gold (Tables 2) is Au+ and Au3+ ions adsorbed in СoFe2O4/Au NPs. Exactly these species (Au+ and Au3+ ions) desorbs during the further cycles of gold deposition (treatment with methionine and other reagents, washing with water, magnetic separation, ultrasonic treatment). However, Au0 nanoparticles are fixed on the NPs and only slightly detached from the magnetic core under the influence of a magnetic field, the repetitive deposition cycles and during post-synthetic processing. There is no contradiction between TEM and XPS data, because TEM can determine only the presence of gold nanoparticles, but not the adsorbed gold ions. XPS determines all gold species on the surface. We agree with the reviewer that our text is somewhat unclear and have made the necessary explanations in the text for clarity.
- As regard the Co and Fe content, I agree with the authors that XPS provide information about the surface composition. In this point I can understand that they don’t have the exact proportion of a stoichiometric compound. However, in my opinion, this does not explain the variation in the Co/Fe ratio among the samples from 1.5 to 0.6 in this new version of the manuscript and between 1.75 and 0.6 in the former version.
Reply: We are thankful for your concern. We checked the XPS data after remove carbon from Table 1 and found typographical errors and corrected the Table 1. However, the variations in the Co/Fe ratio among the samples remain and they are more than the experimental error (20-30%). At the moment, we can’t explain the reasons for the change in the atomic ratio of Co/Fe on the surface of CoFe2O4. We are going to find the answer in the literature and conduct additional research. However, the XRD data before and after the coating (Figure 1, 3, 5) indicate that the phase composition of the samples does not change during the coating (only some Au0 forms) and corresponds to the stoichiometry of cobalt ferrite.
- In this point I realized that there was a part of my question missing. I wrote:
In figure 1 they presented the XRD of the CoFe2O4 particles before the process in which Au is added. It would be interesting to have this information as the authors have already done in previous similar works (i.e. [38]).
When I wanted to ask the following:
In figure 1 they presented the XRD of the CoFe2O4 particles before the process in which Au is added. It would be interesting to have this information AFTER Au ADDITION as the authors have already done in previous similar works (i.e. [38]). This will provide light to this point, to check that the CoFe2O4 nanoparticles are not affected by the treatment to add the Au clusters. Please add XRD of the samples with Au.
Reply: We have added the electron microdiffraction analysis data obtained for all samples to the manuscript as figure 3d-f. We also performed XRD analysis for the sample after 10 gold deposition cycles (Figure 5). The gold deposition doesn't affect the structure and stoichiometry of cobalt ferrite (element concentrations, wt % : Co-24.3, Fe-46.0, O-26.3, Au – 3.3). Before the gold deposition element concentrations, wt % : Co-25.1, Fe-47.6, O-27.2.
- Concerning the binding energy of the C peak, one assumes that if the authors use the BE HOPG is because is the main component of C 1s. It was just a question that I thought it could be a typographical error.
By the way, why do the authors removed not only the binding energies of table 1 but also all reference to the carbon, that is the main specie detected (close to 60% at in all samples) without any mention in the text?. These new values provide a non-realistic vision of the results obtained.
Reply: Indeed, the content of carbon is high but it is due to signals from HOPG support, ligand adsorbed and adventitious carbon contaminations. So, we decided to remove carbon from Table 1 to avoid confusion. The whole compositions can be seen in survey spectra (Figure 7).We agree with Reviewer that this may be misleading and now added a sentence in the text explaining the matter.
- In the core level spectra, the authors still put vertical lines to indicate the components in places where hinders the following of the shoulders of the original data. They must be careful with that. It makes more difficult the evaluation of the results for the reader.
Reply: Please note that figures show the results of fitting, that is, the maxima of the components, and the vertical lines marks their energy positions rather than shoulders. To our knowledge, this is the common for presenting XPS data.
- SQUID measurements: the authors did not include any reference in the text of why they only put the results of one sample (only in the answer of the question).
Reply: We have added reference in text.
- Furthermore, they explained the decrease of Ms by the presence of diamagnetic Au. Au is diamagnetic in bulk. When you go to the nanoscale, you can find a vast literature discussing about induced ferromagnetism an othereffects. What is clear is that Au is weakly polarizable and can contribute to the magnetic response of the system.
Reply: First of all, we are sorry that we made a technical error in the text: the magnetic properties of the samples were examined using not SQUID, but a vibrating sample magnetometer [1, 2]. We have corrected that.
As for the ferromagnetism of gold, as far as we know from the literature [3-5] this effect has mostly been exhibited by gold nanocrystalline films as well as functionalized gold nanoparticles with size less than 5 nm (“all reports that we are aware of deal with functionalized particles or surfaces [4]”). Only a few papers have been published on the ferromagnetic behaviour of separate bare gold nanoclusters [5]. But the obtained results are often contradictory and irreproducible (“magnetic behaviours showed a great variability, for unclear reasons” [4]). A clear understanding of this phenomenon is still missing [3]. Magnetism in gold nanoclusters is thus demonstrated to be the outcome of a very delicate balance of factors. So, the authors [3] claim; “To obtain reproducible results, the samples must be controlled for composition and thus be monodisperse with atomic precision”. We think that we could hardly obtain ferromagnetic gold under the conditions of our experiment with uncoated gold nanoparticles (According to thermogravimetric analysis results (Support Information, Figure 1-2), the amount of organic substances (amino acid methionine and its oxidation products) on the CoFe2O4/Au nanoparticles surface is negligible (2.5 wt. %). Besides, " magnetometry techniques are generally used to study the magnetic properties of materials but have failed to provide coherent results for Au NPs” [4]. In addition, the maximum recorded saturation magnetisation value was about 5 emu/g at liquid helium temperature for gold nanoparticles with a size of 1.9 nm showing ferromagnetic-like behavior [4]. This is at least an order of magnitude less than for СoFe2O4. In most cases, this value for gold nanoparticles is much lower and decreases with increasing temperature. We provide results at 298 K. Therefore, even being ferromagnetic, gold nanoparticles lead to a decrease in magnetisation of the cobalt ferrite NPs. However, we have deleted word “diamagnetic” to avoid confusion.
- Velikanov, D.A. Vibration magnetic meter. RF patent for the invention RU2341810 (C1). Publ. 20.12.2008, Bulletin No. 35.
- Velikanov, D.A. Vibrating magnetometer. RF patent for the invention RU2339965(C1). Publ. 27.11.2008, Bulletin No. 33
- Agrachev, M., Antonello, S., Dainese, T., Ruzzi, M., Zoleo, A., Aprà, E. Govind, N., Fortunelli, A., Sementa, L., Maran, F. Magnetic Ordering in Gold nanoclusters. ACS Omega 2017, 2, 2607−2617. DOI: 10.1021/acsomega.7b00472
- Nealon, G. L., Donnio, B., Greget, R., Kappler, J.-P., Terazzi, E., Gallani, J.-L. Magnetism in gold nanoparticles. Nanoscale, 2012, 4, 5244–5258. DOI: 1039/c2nr30640a
- Venäläinen, A., Jalkanen, P., Tuboltsev, V., Meinander, K., Savin, A., Räisänen, J. Ferromagnetism in bare gold nanoagglomerates produced by nanocluster deposition. Journal of Magnetism and Magnetic Materials, 2018б 454, 57 – 60. DOI: 10.1016/j.jmmm.2018.01.041

Reviewer 2 Report
Authors replied to most of my concerns, and I understand the situation that they could not perform te new required experiments due to the pandemic.
I consider the work can be published.
Best regards,
***
Author Response
We thank the reviewer for spending his precious time to review this article.
Reviewer 4 Report
I think this paper can be accepted.
Author Response

(The authors gave the same response as above.)
